



# Evaluation of four years continuous $\delta^{13}C(CO_2)$ data using a running Keeling approach

Sanam Noreen Vardag, Samuel Hammer, and Ingeborg Levin

Institut für Umweltphysik, Heidelberg University, Germany

*Correspondence to:* S. N. Vardag (svardag@iup.uni-heidelberg.de)

**Abstract.** As different carbon dioxide ($CO_2$) emitters have different carbon isotope ratios, measurements of atmospheric $\delta^{13}C(CO_2)$ and $CO_2$ concentration contain information on the $CO_2$ source mix in the catchment area of an atmospheric measurement site. Often, this information is illustratively presented as mean isotopic source signature. Recently an increasing number of continuous measurements of $\delta^{13}C(CO_2)$ and $CO_2$ have become available, opening the door to quantification of $CO_2$

shares from different sources at high temporal resolution. Here, we present a method to compute the $CO_2$ source signature ($\delta_S$) continuously without introducing biases and evaluate our result using model data. We find that biases in $\delta_S$ are smaller than 0.2 ‰ with uncertainties of about 1.2 ‰ for hourly data. Applying the method to a four year data set of $CO_2$ and $\delta^{13}C(CO_2)$ measured in Heidelberg, Germany, yields a distinct seasonal cycle of $\delta_S$. Disentangling this seasonal source signature into its source components is, however, only possible if the isotopic end members of these sources, i.e., the biosphere, $\delta_{bio}$, and the

fuel mix, $\delta_F$, are known. From the mean source signature record in 2012, $\delta_{bio}$ could be reliably estimated only for summer to (-25 ± 1) ‰ and $\delta_F$ only for winter to (-32.5 ± 2.5) ‰. As the isotopic end members $\delta_{bio}$ and $\delta_F$ were shown to change over the season, no year-round estimation of the fossil fuel or biosphere share is possible from the measured mean source signature record without additional information from emission inventories or other tracer measurements, such as $\Delta^{14}C(CO_2)$.

## 1 Introduction

A profound understanding of the carbon cycle requires closing the atmospheric $CO_2$ budget at regional and global scale. For this purpose it is necessary to distinguish between $CO_2$ contributions from oceanic, biospheric and anthropogenic sources and sinks. Monitoring these $CO_2$ contributions separately is desirable for improving process understanding, investigating climatic feedbacks on the carbon cycle and also to verify emission reductions and designing $CO_2$ mitigation strategies (Marland et al., 2003; Gurney et al., 2009; Ballantyne et al., 2010). A possibility to distinguish between different $CO_2$ sources and sinks

utilizes concurrent $^{12}CO_2$ and $^{13}CO_2$ observations in the atmosphere. The carbon isotope ratio can be used to identify and even quantify different $CO_2$ emitters if every emitter has its specific known $\delta^{13}CO_2$ signature. For example, the $CO_2$ fluxes from land and ocean can be distinguished using the ratio of stable carbon isotopologue $^{13}CO_2/^{12}CO_2$ in addition to $CO_2$ concentration measurements (Mook et al., 1983; Ciais et al., 1995; Alden et al., 2010). In other studies, measurements of $^{13}CO_2$ have been used to distinguish between different fuel types (Pataki, 2003; Lopez et al. 2013) or to detect ecosystem

behavior (Torn et al., 2011), giving only a few examples of the many published in the literature.





In the last decade, new optical instrumentation has been developed, simplifying continuous isotopologue measurements. This led to an increasing deployment of these instruments, therby increasing the temporal and spatial resolution of $^{13}$C(CO$_2$) and CO$_2$ data (Bowling et al., 2003; Tuzson et al., 2008; McManus et al., 2010; Griffith et al., 2012; Vogel et al., 2013; Vardag et al., 2015a, Eyer et al., 2016). These data records may lead to an improved understanding of regional CO$_2$ fluxes allowing

estimates of mean $\delta^{13}$C source signatures at high temporal resolution. Estimating mean source signatures from concurrent $\delta^{13}$C(CO$_2$) and CO$_2$ records over time provides e.g. insight into temporal changes in the signatures of two different CO$_2$ sources such as fossil fuels and the biosphere, if their relative share to the CO$_2$ offset is known. This may e.g. give insight into biospheric responses to climatic variations like drought, heat, floods, vapor pressure etc. (Ballantyne et al., 2010; Ballantyne et al., 2011; Bastos et al., 2016). Likewise, the mean source signature can be used to separate between different source CO$_2$

contributions, if the isotopic end members of these sources are known at all times (Pataki, 2003; Torn et al., 2011; Lopez et al. 2013; Newman et al., 2015).

Many studies have successfully used the Keeling- or Miller-Tans- approach (Keeling, 1958, 1961; Miller and Tans, 2003) to determine source signatures in specific settings (e.g. Pataki, 2003; Ogée et al., 2004; Lai et al., 2004; Knohl et al., 2005; Karlsson et al., 2007; Ballantyne et al., 2010). However, the situations in which Keeling and Miller-Tans plots yield correct

results need to be selected carefully (Miller and Tans, 2003). Only if all possible pitfalls are precluded, the Keeling intercept (or the Miller-Tans slope) can be interpreted as gross flux-weighted mean isotopic signature of all CO$_2$ sources and sinks in the catchment area of the measurement site. Especially in polluted areas with variable source/sink distribution, estimation of isotopic signature using a Keeling- or Miller-Tans-plot requires a solid procedure, e.g. accounting for wind direction changes or simultaneously occurring CO$_2$ sinks and sources. In this study, we discuss the possible pitfalls of CO$_2$ source signature

determination from a continuous data set using the Keeling approach and follow a specific modification of this method for automatic and bias-free mean source signature determination. We test this method with model-simulated CO$_2$ mole fraction and $\delta^{13}$C(CO$_2$) data. Using a modeled data set where all source signatures are known, enables us to check if the calculated source signature is correct, which is vital when evaluating measured data with an automated routine. Having found a method to determine the isotopic signature of the mean source signature correctly from measured CO$_2$ and $\delta^{13}$C(CO$_2$) data, we discuss,

which information can be reliably extracted from these results.

## 2   Determination of source signature

### 2.1   Classical Keeling and Miller-Tans approach

Keeling (1958, 1961) showed that the mean isotopic signature of a source mix can be calculated by re-arranging the mass balance of total CO$_2$

$$CO_{2tot} = CO_{2bg} + CO_{2S} \tag{1}$$

and of $\delta^{13}$C of total CO$_2$, i.e. $\delta_{tot}$:

$$\delta_{tot} \cdot CO_{2tot} = \delta_{bg} \cdot CO_{2bg} + \delta_S \cdot CO_{2S} \tag{2}$$





to:

$$\delta_{tot} \approx CO_{2bg}/CO_{2tot} \cdot (\delta_{bg} - \delta_S) + \delta_S \qquad (3)$$

where $CO_{2bg}$ and $\delta_{bg}$ are the concentration and $\delta^{13}C(CO_2)$ of the background component and $CO_{2S}$ and $\delta_S$ are the concentration and $\delta^{13}C(CO_2)$ of the mean source, respectively. In a graphical evaluation when plotting $\delta_{tot}$ versus $1/CO_{2tot}$, this yields
$\delta_S$ as the $\delta$-intercept of the regression of all measurement points (cf. Fig.1a).

Miller and Tans (2003) have suggested an alternative approach to determine the mean isotopic signature by re-arranging Eqs. 1 and 2 such that $\delta_S$ is the regression slope when plotting $CO_{2tot} \cdot \delta_{tot}$ versus $CO_{2tot}$:

$$CO_{2tot} \cdot \delta_{tot} = \delta_S \cdot CO_{2tot} - CO_{2bg}(\delta_{bg} - \delta_S) \qquad (4)$$

They argue that this approach might be advantageous since the isotopic signature does not need to be determined from extrap-
olation to $1/CO_2=0$, which could introduce large errors in the $\delta_S$ estimate. Zobitz et al. (2006) have compared the Keeling and the Miller-Tans approach (Eqs. 3 and 4) and found no significant differences between both approaches when applied to typical ambient $CO_2$ variations. We were able to reproduce this result with our model-simulated data set (cf. Sect. 3.2). Differences between both approaches were (0.00 ± 0.04) ‰ when applying certain criteria (standard deviation of intercept < 2 ‰, $CO_2$ range within 5 hours >5 ppm), which will be motivated in Sect. 2.3. In our study, we use a Keeling plot for calculation of the
mean source signature, but using a Miller-Tans plot seems just as good. Note that the isotopic signature of the mean source $\delta_S$ can be determined from linear regression without requiring a background $CO_2$ and $\delta^{13}C(CO_2)$ value. However, the Keeling and Miller-Tans approaches are only valid if the background and the isotopic signature of the source mix $\delta_S$ are constant during the period investigated (Keeling, 1958, Miller and Tans, 2003). Further, the approaches are only valid when sources and sinks do not occur simultaneously. Miller and Tans (2003) gave an example, which showed that as soon as sources and sinks of
different isotopic signature/fractionation occur simultaneously, the determination of isotopic signature of the source/sink mix is not *per se* possible. In these cases, the results cannot be interpreted as mean flux-weighted source signature anymore. This has very unfortunate consequences, since in principle we are interested in determining the isotopic signature of the source mix of a region during all times, i.e. also during the day when photosynthesis cannot be neglected. Pataki (2003), Miller and Tans (2003) and Zobitz et al. (2006) compared different fitting algorithms for the regression and came to different recommendations.
Orthogonal distance regression (ODR) and weighted total least squares fits (WTLS) (model 2 fits) take into account errors on x and y, whereas ordinary least squares (OLS) minimization (model 1 fit) only takes into account y-errors. Zobitz et al. (2006) have found differences between both fitting algorithms especially at small $CO_2$ ranges. We have also applied a model 1 (OLS) and model 2 (WTLS) fit to our simulated data and have not found any significant differences ((0.00 ± 0.01) ‰) between them when applying certain criteria (error of intercept < 2 ‰, $CO_2$ range within 5 hours >5 ppm, see Sect. 2.2). In this study,
however, we use a WTLS-fit for the determination of the intercept and its uncertainty.

## 2.2 Running Keeling approach

For a continuous long-term data set, we suggest an automatic routine to determine the mean isotopic signature of the source mix. We call this approach the "running" Keeling approach. It is similar to the moving Keeling plot for $CH_4$ currently suggested




by Röckmann et al. (2016). In our case of $CO_2$ we also have to take into account the possibility of simultaneously occuring sinks and sources, which is not important in the case of $CH_4$. Our running Keeling approach is a specific case of the classical Keeling approach (Eq. 3) (Keeling, 1961) as it uses only five hourly-averaged measurement points of $CO_2$ and $\delta^{13}C(CO_2)$ fitting a regression line through these five data points (cf. Fig. 1a, illustrated only for three data points for clarity of inspection).

We choose 5 hours as a compromise of maximum number of data points in a minimizing period, in which the source mix does not change significantly. No background value is included in the regression. The running Keeling approach works such that, e.g. for the determination of the mean source signature at 3 pm, we use the hourly $CO_2$ and $\delta^{13}C(CO_2)$ measurements from 1 pm to 5 pm and fit a regression line. Next, for the determination of the source signature at 4 pm, we use the $CO_2$ and $\delta^{13}C(CO_2)$ measurements from 2 pm to 6 pm and so on.

## 2.3  Filter criteria of the running Keeling approach

In order to prevent pitfalls in the regression-based determination of mean isotopic signature, we set a few criteria for the running Keeling plots to "filter" out situations, in which a Keeling plot cannot be performed. These filter criteria are also similar in type to the ones introduced by Röckmann et al. (2016). We here explain why these filter criteria are needed for $CO_2$ and how they are set. A prerequisite for the Keeling plot is that the source mix as well as the background need to stay constant during the

investigated period (see Fig. 1a). Varying source mixes may occur when e.g. the wind direction and therewith the footprint of the measurement site change, or if the emission patterns themselves change over time. This may lead to strong biases of the regression-based mean isotopic source signature (illustrated in Fig. 1b). We eliminate these cases by inspecting the error of the determined intercept $\delta_S$. If the source mix or the background significantly change within five hours, the data points will not fall on a straight line and the error of the intercept will increase. We here set an error of 2 ‰ (in a WTLS fit) as threshold between

an acceptable and a "bad" fit, after having inspected many Keeling plots individually. Also, we demand a monotonous increase of $CO_2$ within 5 hours, as a decrease of would be due to either a sink of $CO_2$ or a breakdown of the boundary layer inversion associated with a change of catchment area of the measurement, both biasing the resulting mean source signature.

As mentioned before, the determination of a mean isotopic signature is not *per se* possible during the day when $CO_2$ sinks and sources are likely to occur simultaneously (Miller and Tans, 2003). This can be explained in the Keeling plot by the vector

addition of $CO_2$ source and sink mixing lines with different isotopic signatures, resulting in a vector with an intercept different from the expected one, leading to an isotopic signature, which can even lie outside the expected range of the isotopic source end members (see Fig. 1c). This potential bias is stronger, the smaller the net $CO_2$ signal is. Therefore, e.g. for evaluation of the Heidelberg data, we demand an increase in $CO_2$ during the 5 hour period of at least 5 ppm to exclude periods where the photosynthetic sink is similarly strong as $CO_2$ sources. This normally leads to an exclusion of daytime periods, when the

boundary layer inversion typically breaks up and the photosynthetic sink is most pronounced. During winter, it may happen that the inversion does not break up due to the cold surface temperatures, but in this season, photosynthetic activity is typically much smaller than fossil fuel emissions and therefore biases of the regression-based mean source signature are only small. In the next section, we show that with these filter criteria, which we chose empirically, we are able to successfully remove those



source signatures, where the underlying assumptions for the Keeling approach are not met. In Sect. 3.2, we will also briefly discuss how sensitive the result is to the choice of filter criteria.

# 3 Test of the running Keeling approach with modeled data

We apply the running Keeling method to a modeled $CO_2$ and $\delta^{13}C(CO_2)$ data set. As also pointed out by Röckmann et al. (2016)
in their $CH_4$ study, this has the advantage that we can test and evaluate our filter criteria as we know exactly the individual isotopic source signatures that created the modeled data set and thus, the contribution-weighted mean isotopic source signature at every point in time. Details on the STILT model and on the computation of the modelled $CO_2$ and $\delta^{13}C(CO_2)$ record as well as of the resulting mean source signature, $\delta_S^{STILT}$, are given in Appendix A.

## 3.1 Filter criteria of modeled source signature

We apply the same filter criteria to the calculated mean source signature of the STILT modelled data set $\delta_S^{STILT}$, as to the regression-based mean source signature (Sect. 2.3). The "unfiltered" source signatures (black in Fig. 2a) are 0-2‰ more enriched than the "filtered" source signatures (blue). This offset is mainly caused by the daytime source signatures, which are on average more enriched than nighttime source signatures (Fig. 2b), but more likely to be filtered out based on the criteria of Sect. 2.3.

## 15 3.2 Evaluation of running Keeling approach

We can now evaluate the running Keeling method and the filter criteria based on the model data and test if they allow a bias-free retrieval of the mean source signature. In Fig. 3a, we compare the regression-based source signatures to the filtered reference source signature of Fig 2a, which we have extracted from the model. We do not only compare the mean difference of the mean source signature, but the hourly differences of the mean source signature as well as the smoothed difference. This enables us
to clearly state how well we are able to determine the hourly mean source signature and its long-term trend.

Fig. 3a displays the filtered seasonal changes of the source signature for the year 2012. The running Keeling approach is able to extract the seasonal variability of the mean isotopic signature correctly. The median difference (and inter-quartile range) between smoothed regression-based (red) and smoothed modeled (blue) approach (both smoothed with 50% percentile filter with window size of 100 hours, no smoothing 50 points in front of large data gaps) is 0.0 ± 0.4 ‰. On a shorter diurnal time
scale, we also compare individual hourly results for the source signature (stars in Fig. 3b,c). The inter-quartile range of the filtered hourly difference between both the reference $\delta_S^{STILT}$ and the running Keeling signature is ca. 1.2 ‰ throughout the year, but the median difference is small (0.2 ‰). The source signature of the model reference and running Keeling source signature show the same temporal pattern both, in summer and in winter. Further, we find that if we do not apply all of the criteria described in Sect. 2.2 (unfiltered data in Fig. 3b,c), we see larger differences between regression-based source signature
(from the running Keeling plot) and the STILT reference values. Note, however, that with the criteria established in Sect. 2.3, we have rejected about 85% of all estimated source signatures. Depending on the application, it may be worthwhile to loosen



the filter criteria to increase the data coverage. For example, if one sets no criteria for the minimal $CO_2$ range, but only for the error of the slope ($< 2‰$), about $60\%$ of all data remain for the estimated source signature, but the median difference between model- and Keeling-based results increases to $0.3‰$ and the interquartile range increases to $2.4‰$ (hourly data), which is about twice of what we found before. Withdrawing all filter criteria, but using only night time values, leads to a coverage of about

$35\%$ (night time) and an interquartile range of $3.5‰$. The filter criteria, which we use here (Sect. 2.3) are, thus, rather strict, but we are confident to precisely extract the correct source signature from the $\delta^{13}C(CO_2)$ and $CO_2$ record at highest temporal resolution.

## 4    Application of the running Keeling approach

### 4.1    The measured source signature record in Heidelberg

We now apply this approach to real measured data. We use the Heidelberg $CO_2$ and $\delta^{13}C(CO_2)$ record on hourly time resolution (Fig. B1) to compute the isotopic source signature via regression (Fig. 4). The quality of the Heidelberg $CO_2$ and $\delta^{13}C(CO_2)$ record is assessed in the Appendix B. We observe a distinct seasonal cycle of the mean isotopic source signature in Heidelberg. Smoothed minimum values of about -32 ‰ are reached in winter. Maximum values of about -26 ‰ occur in summer. This principal pattern is reproduced every year. Additionally, the first year shows a more enriched summer maximum

source signature. A number of data points (less than $0.5\%$) lie outside the range of realistic end members between -20 and -45 ‰ of any source in the catchment area (see Table 1). These outliers can be explained statistically by the uncertainty of the running Keeling approach. From the model analysis, we expect the inter-quartile range of $\delta_S$ for the Heidelberg catchment area to be about $1.2‰$, in accordance to Fig. 4 ($1.8‰$). Our record of the mean source signature in Heidelberg provides a first insight into the source characteristics at the measurement station. It reaches its minimum in winter when we expect residential

heating (mainly isotopically depleted natural gas, see Tab. 1) to contribute significantly to the source mix. The source signature reaches its maximum in summer when more enriched biospheric fluxes are expected to dominate the $CO_2$ signal. This observed seasonal cycle in Heidelberg (Fig. 4) is very similar to the filtered modelled source signature (Fig. 3a) in amplitude as well as phase.

### 4.2    Extracting information on the isotopic end members $\delta_{bio}$ and $\delta_F$ from $\delta_S$

We now want to elaborate what quantitative information can be drawn from the mean source signature record in Heidelberg about its components. Details on the Heidelberg measurement site and catchment area can be found in Vogel et al. (2010).

#### 4.2.1    Formulation of question

For a continental measurement site such as Heidelberg, we have to assume that there are at least two main source types of $CO_2$ in the catchment area: Fuel $CO_2$ and $CO_2$ from the biosphere. In this simplest case, we essentially have one equation ($\delta_S$, Eq.



6) with three unknown variables ($\delta_{bio}$, $\delta_F$ and the fuel (or biosphere) share $f_F$) and only if two of these variables are known, the third variable can be quantified from the measurements:

$$\delta_S = \frac{CO_{2F}}{\Delta CO_2} \cdot \delta_F + \frac{\Delta CO_2 - CO_{2F}}{\Delta CO_2} \cdot \delta_{bio} \qquad (5)$$

$$= f_F \cdot \delta_F + (1 - f_F) \cdot \delta_{bio} \qquad (6)$$

Which of the variables is the one to be estimated depends, of course, on the research question. If the fossil fuel share and end members are well known from inventories, one could be especially interested in determining the isotopic end member $\delta_{bio}$ in order to study biospheric processes and their feedback to climatic parameters (Ciais et al., 2005; Ballantyne et al., 2010; Salmon et al., 2011). Contrary, one may be interested in determining the relative share of fossil fuel $CO_2$ in the catchment area (with known $\delta_{bio}$ and $\delta_F$) to monitor emission changes independently from emission inventories. In our discussion, we focus
on the determination of the fossil fuel share, but the arguments for most parts are analog for other research questions.

As noted, a quantification of the relative shares of fossil fuel and the biospheric $CO_2$ at continental stations is only possible if information on the isotopic end members of both source categories are available. For example, Vardag et al. (2015b) used the isotopic signatures of $\delta_{bio}$ (assumed to be known within a fixed uncertainty) and $\delta_F$ (obtained by calibration with $\Delta^{14}C(CO_2)$) to calculate the fossil fuel $CO_2$ contribution from the (continuously) measured $CO_2$ and $\delta^{13}C(CO_2)$ signal. However, knowing
the isotopic signatures $\delta_{bio}$ and $\delta_F$ over the entire course of the year, requires an extensive number of measurements at the relevant sources throughout the year and further assumptions how to extrapolate the source signature of the point measurements to a mean source signature of all relevant sources. Therefore, we ask here, if we can obtain information on these end members from our measured source signature record, despite the fact that we have three unknown variables and only one equation. In the following, we discuss this question exemplary for the year 2012, for which we have modeled data, inventory information
and an almost complete measurement record.

### 4.2.2 One source approximation

In general, in order to obtain information on $\delta_{bio}$ ($\delta_F$), we require information on the fuel $CO_2$ share and $\delta_F$ (on the fuel $CO_2$ share and $\delta_{bio}$). However, in cases where the relative share of the biosphere (fossil fuels) is negligible, the isotopic signature of $\delta_F$ ($\delta_{bio}$) would equal the mean isotopic signature. In these cases, the number of unknown variables would be reduced to one
as the fossil fuel (biospheric) share is $\approx 100\%$ and $\delta_{bio}$ ($\delta_F$) does not contribute significantly to the mean source signature. In a typical catchment area, the relative share of fossil fuels and of the biosphere will not be negligible throughout the year, but in winter, fossil fuel $CO_2$ will dominate while in summer the biospheric $CO_2$ will dominate the $CO_2$ offset compared to the background. E.g. from the STILT model results for Heidelberg (Sect. 3.2 and Appendix A), we perceive that on cold winter days in Heidelberg, the fossil fuel share can be about 90 to 95% of the total $CO_2$ offset. In summer, it reaches a minimum
at about 20%. We may, thus, be able to obtain information about the isotopic end members of $\delta_F$ in winter ($\delta_{bio}$ in summer), when the mean source signature is dominated by the fossil fuel (biospheric) share.





To calculate the resulting isotopic end members of $\delta_i$ from the measured source signature (and with that to solve Eq. 6), we require the fossil fuel $CO_2$ share from STILT and the bottom-up emission inventory EDGAR. However, as we only require the share and not the absolute concentration, we are largely independent from potentially large model transport errors. We assume an absolute uncertainty of 10 % of the fossil fuel share (and of the biospheric share respectively).

To determine $\delta_F$ in addition to the fuel $CO_2$ share, we require a value for $\delta_{bio}$. Here we use a typical mean value of the isotopic end member of $\delta_{bio}$= -25 ‰ and assume a seasonal cycle as determined for Europe by Ballantyne et al. (2011) (see Fig. 2 and 3 in Ballantyne et al. (2011)) displayed in Fig. 5a as solid green line. We show $\delta_{bio}$ with two possible uncertainties of 0.5 and 2 ‰. As expected, the uncertainty of the unknown $\delta_F$ is only acceptably small when the relative share of the biosphere becomes negligible, which is the case in winter (Fig. 5a). The isotopic end member of $\delta_F$ in winter is about (-31 ± 2.5) ‰ in
January to March 2012 and decreases to (-32.5 ± 2.5) ‰ in November to December 2012. Further, Fig. 5a shows that the best estimate of the resulting isotopic signature $\delta_F$ is more depleted in summer than in winter. This curvature is opposite from what we would expect from EDGAR (2010) transported by STILT (see assumed $\delta_F$ in Fig. 5b). Only when assuming an uncertainty of the biospheric end member of $\pm$ 2 ‰ or more, the uncertainty range of the estimated $\delta_F$ allows more enriched $\delta_F$ signature in summer than in winter. This suggests that the isotopic source signature of the biosphere in summer is most probably more
depleted (by about 2 ‰) than the previously assumed $\delta_{bio}$ value based on Ballantyne et al. (2011).

To estimate $\delta_{bio}$ (Fig. 5b), we require (besides the fossil fuel share) the isotopic source signature $\delta_F$. Here we use $\delta_F$ calculated with the STILT model on the basis of EDGAR emissions and source signatures according to Tab. 1. Its annual mean value is -31 ‰ and it shows a seasonal cycle with more enriched signatures in summer than in winter. We show the results for $\delta_{bio}$ for two possible $\delta_F$ uncertainties of 1 and 3 ‰ (see Fig. 5b). The best-estimate of the isotopic end member of $\delta_{bio}$
in summer is about -25.0 ± 1 ‰ in June to August 2012. This reinforces the presumption that $\delta_{bio}$ is more depleted than the assumed $\delta_{bio}$ value based on Ballantyne et al. (2011) during summer.

### 4.2.3  Evaluation of the precision

The uncertainty of the isotopic end members in Fig. 5a and b has three components: 1) The uncertainty of the fossil fuel $CO_2$ share estimated from STILT, which we assume to be about 10% (absolute) in our case, 2) the uncertainty of the other known
isotopic end member (0.5 and 2 ‰ for $\delta_{bio}$ or 1 and 3 ‰ for $\delta_F$) and 3) the uncertainty of the measured mean source signature itself (ca. 0.5 ‰, see Sect. 3.2). Note, that an uncertainty of 10% of the fossil fuel share is at the low end of uncertainties. However, an uncertainty of 20% of the fossil fuel share would increase the uncertainty in the unknown isotopic end members by only 0.2 - 0.4 ‰ for $\delta_{bio}$ in summer and $\delta_F$ in winter, respectively.

The derived uncertainty of $\delta_F$ in winter is about 2.5 ‰ and that of $\delta_{bio}$ in summer is about 1.5 ‰. An uncertainty of ± 2.5
‰ for $\delta_F$ is rather large if we want to use this observation-based top-down result for further quantitative source apportionment. Vardag et al. (2015b) showed that a misassignment of 2.5 ‰ in $\delta_F$ leads to a bias in the continuous fuel $CO_2$ estimate of about 15% for an urban measurement site like Heidelberg. The observation-based biospheric end member $\delta_{bio}$ has an uncertainty of only about 1 ‰ in June to August 2012, which is a very well constraint value for this period.



### 4.2.4 Evaluation of accuracy

If both isotopic end members stayed constant over the course of one year, we would now be able to actually estimate the fossil fuel $CO_2$ share (and its uncertainty) continuously throughout the year without requiring any additional information, such as inventories or $\Delta^{14}C(CO_2)$ for calculation of $\delta_F$ from the mean source signature. However, from bottom-up information, we

would neither expect the isotopic value of the biosphere nor that of the fossil fuel mix to remain constant throughout the year. In contrary, we would expect the biosphere to show a distinct seasonal pattern e.g. due to the change in fraction of respiration from C3/C4 plants over the course of the year or influences of climatic conditions on biospheric respiration (e.g. Still et al., 2003; Ciais et al., 2005). A seasonal cycle of $\delta_F$ is also expected with more enriched values in summer, when the contribution of residential heating (and therewith of depleted natural gas) is much smaller than in winter. Therefore, if we have varying

isotopic end members of $\delta_F$ and $\delta_{bio}$, we cannot estimate the fossil fuel share correctly for the entire year. But, if the amplitude of these changes is small, the biases in fossil fuel $CO_2$ will be small as well. Vardag et al. (2015b) have shown that from a limited number of $^{14}C(CO_2)$ grab samples distributed over the year, the true annual mean value of $\delta_F$ can be obtained. Here we show that from the mean $\delta^{13}C$ source signature only a reliable winter value is obtained, potentially introducing summer biases (as well as annual averaged biases) into the fuel $CO_2$.

### 4.2.5 Possible strategy to obtain $\delta_F$ and $\delta_{bio}$

To determine $\delta_{bio}$, one can take the summer value of $\delta_{bio}$ from the source signature record following Sect. 4.2.2. As no reliable determination of $\delta_{bio}$ is possible during the rest of the year based only on atmospheric observations, there must be either very good bottom-up literature values for the catchment area of interest or frequent measurement campaigns at the sources must be performed. However, the disadvantage of using a bottom-up approach is that usually only information from few specific

sites are available, which need then to be upscaled correctly such that they are representative of the entire catchment area. For a determination of $\delta_F$ in the entire year, one can use $\Delta^{14}C(CO_2)$ measurements (following Vardag et al. (2015b)) or rely on the bottom-up inventory information. To obtain correct source signatures of the different fossil fuel categories, measurements close to these sources are required to support or refute the inventory-model based estimates. These measurements again need to be upscaled correctly.

## 5 Conclusions

Many measurement stations are currently being equipped with new optical instruments, which measure $\delta^{13}C(CO_2)$ aiming at a more quantitative understanding of the carbon fluxes in their catchment area. If this additional $\delta^{13}C(CO_2)$ data stream is not directly digested in regional model calculations, the mean isotopic source signature is often computed from the $\delta^{13}C(CO_2)$ and $CO_2$ records for the analysis of the source composition. Essentially, this source signature provides the same degree of

information as the measured $\delta^{13}C$ and $CO_2$ records themselves, but is a more intuitive and therefore common form for further interpretations.





We re-emphasize here that a bias-free determination of source signature requires carefully selecting the data for situations, in which determination of source signature with a Keeling plot can provide reliable results. This excludes periods, when sinks and sources occur simultaneously, when the source mix changes or when the signal-to-noise ratio is too low (Keeling, 1958; Keeling, 1961; Miller and Tans, 2003). We therefore developed filter criteria and show that the routine and accurate

determination of $^{13}$C(CO$_2$) source signature is possible, if the introduced filter criteria are applied. As suggested by Röckmann et al. (2016), we use a modeled data set for validation of the approach. We find that for a station like Heidelberg, the bias introduced by our analysis is only $(0.2 \pm 1.2)$ ‰ for hourly data. The uncertainty decreases in the long-term to $(0.0 \pm 0.4)$ ‰. We are therefore able to estimate the source signature correctly. However, as the filter criteria are such that the source signatures are more likely to be filtered out during the day than during the night, the long-term source signature is not representative of

real daily averages, but only of periods, where the data was not filtered out (mainly nighttime). This problem does not occur for CH$_4$, which has only weak daytime sinks.

By applying the running Keeling plot procedure to a real measured data set in Heidelberg, we are able to determine the source signature over the course of four years. We find a distinct seasonal cycle of the mean source signature with values of about -26 ‰ in summer and about -32 ‰ in winter. This general behavior was expected due to the larger relative contribution

of more depleted fossil fuel CO$_2$ in winter. For a unique interpretation of the mean source signature, possible sources in the catchment area need to be identified. As soon as there are more than one source, the source signature is a function of the isotopic end members of all sources, as well as of their relative shares. Therefore, to study the seasonal and diurnal changes of fossil fuel shares at a continental station, information on the isotopic end members of the fossil fuel mix as well as of the biosphere are required on the same time resolution. Unfortunately, the isotopic end members are often not known with high

accuracy. The uncertainty of the isotopic end members often impedes or even prevents a unique straightforward determination of the source contribution in the catchment area (e.g. (Pataki, 2003; Torn et al., 2011, Lopez et al. 2013; Röckmann et al., 2016) and calls for elaborated statistical models based on Bayesian statistics . This important fact is sometimes mentioned, but the consequences for quantitative evaluations are rarely emphasized, preserving the high expectations associated with isotope measurements.

We showed that for the urban site Heidelberg, we can use the observation-based mean source signature record to estimate the isotopic end member $\delta_F$ in winter and the isotopic end member $\delta_{bio}$ in summer within the uncertainties of $\pm 2.5$ ‰ and $\pm 1$ ‰, respectively, when assuming an uncertainty of $\pm 10$ % for the fossil fuel and biospheric CO$_2$ share and an uncertainty of the other isotopic end member $\delta_F$ of $\pm 3$ ‰ and $\delta_{bio}$ of $\pm 2$ ‰. However, in the winter season we cannot obtain any reliable information on $\delta_{bio}$ and in summer we cannot study $\delta_F$. If the isotopic end members would not change within seasons, it would

be possible to determine these constant isotopic signature from our obtained estimates. However, this is not a valid assumption.

Finally, we could show, that even though it is not possible to determine the isotopic end members throughout the year, it is possible to refute certain literature values. E.g. a respiration signature of -23 ‰ in August and September 2012 as reported by Ballantyne et al. (2011) is most likely too enriched as this would lead to more depleted $\delta_F$ values in summer than in winter, which is in contrast to what we would expect based on emission inventories.



## Appendix A: The STILT model

We use the Stochastic Time Inverted Lagrangian Transport (STILT) model (Lin et al., 2003) to evaluate our running Keeling approach. The STILT model computes the $CO_2$ mole fraction by time-inverting meteorological fields and tracing particles emitted at the measurement location back in time to identify where the air parcel originated. This so-called footprint area

is then multiplied by the surface emissions in the footprint to obtain the $CO_2$ concentration at the site in question. Photosynthesis and respiration $CO_2$ fluxes are taken from the vegetation photosynthesis and respiration model (VPRM, Mahadevan et al., 2008). Anthropogenic emissions are taken from EDGARv4.3 emission inventory (EC-JRC/PBL, 2015) for the base year 2010 and further extrapolated to the year 2012 using the BP statistical review of World Energy 2014 (available at: http://www.bp.com/en/global/corporate/about-bp/energy-economics/statistical-review-of-world-energy.html). Additionally,

we use seasonal, weekly and daily time factors for different emission categories (Denier van der Gon et al., 2011). Since the EDGAR inventory is separated into different fuel types, we obtain a $CO_2$ record for each fuel type as well as for respiration and photosynthesis. This allows us, to construct a corresponding $\delta^{13}C(CO_2)$ record by multiplying the isotopic signature of every emission group i to its respective $CO_2$ mole fraction $\delta^{13}C(CO_2)_i \cdot CO_{2,i}$ (see Tab. 1), adding these to a far-field boundary value of $\delta^{13}C(CO_2) \cdot CO_2$ and dividing it by the total $CO_2$ at the model site. The $CO_2$ far-field boundary value for STILT is

the concentration at the European domain border (16°W to 36°E and from 32°N to 74°N) at the position where the backwards traced particles leave the domain. The concentration at the domain border is taken from analyzed $CO_2$ fields generated with TM3 (Heimann and Körner, 2003) based on optimized fluxes (Rödenbeck, 2005). The isotopic boundary value is then constructed artificially by fitting the linear regression between $CO_2$ and $\delta^{13}C(CO_2)$ in Mace Head (year 2011 from World Data Center for Greenhouse Gases, (Dlugokencky et al., 2015)) and applying the function of the regression to the boundary $CO_2$

values in the model. Since, in reality, we also have measurement uncertainties of $CO_2$ and $\delta^{13}C(CO_2)$ we also include a random measurement uncertainty of 0.05 ppm and 0.05 ‰, respectively to the modeled data sets. The $CO_2$ and $\delta^{13}C(CO_2)$ records are used to calculate the regression-based mean source signature following the running Keeling approach (Sect. 2.2).

### A1 Computation of mean modeled source signature

For the reference modeled mean source signature we use a "running" background. In particular, we chose the minimum $CO_2$

value within 5 hours centered around the measurement point as the background value and all contributions from fuel $CO_2$ ($c_{F,i}$), respiration ($c_{resp}$) and from photosynthesis ($c_{photo}$) are computed as offsets relative to the background ($c_{bg}$). This is then comparable to the regression-based running Keeling approach as the lowest and highest $CO_2$ values within five hours span the Keeling plot. We are then able to define and compute the reference modeled mean source signature as:

$$\delta_S^{STILT} = \frac{\sum_i \delta_{F,i}|c_{F,i}| + \delta_{resp}|c_{resp}| + \delta_{photo}|c_{photo}|}{\sum_i |c_{F,i}| + |c_{resp}| + |c_{photo}|} \tag{A1}$$

Note that we use absolute values of all contributions since photosynthetic contributions ($c_{photo}$) are generally negative while source contributions ($c_{resp}$ and $c_{F,i}$) are generally positive, but both should lead to a negative source signature in a Keeling plot. The calculated source signature $\delta_S^{STILT}$ (from Eq. A1) can be seen in Fig. 2a (blue). If we would not take into account



the different signs of respiration and photosynthesis, we would construct isotopic signatures, which are counter-intuitive and not interpretable as mean source signature (Miller and Tans, 2003) as the denominator could converge against zero. When calculating the isotopic source following Eq. A1, we can interpret $\delta_S^{STILT}$ as gross flux weighted mean isotopic signature of sources and sinks.

## 5  Appendix B:  $CO_2$ and $\delta^{13}C(CO_2)$ measurements in Heidelberg

A necessary prerequisite of determining the mean source signature correctly at a measurement site is a good quality of $CO_2$ and $\delta^{13}C(CO_2)$ measurements. Therefore, we briefly describe here the instrumental set-up in Heidelberg, assess the precision of the $CO_2$ and $\delta^{13}C(CO_2)$ measurements and finally present our four years' ambient air record of $CO_2$ and $\delta^{13}C(CO_2)$ in Heidelberg.

### 10  B1   Instrumental set-up and intermediate measurement precision

Since April 2011, atmospheric trace gas mole fractions are measured with an *in-situ* Fourier Transform-InfraRed (FTIR) spectrometer at three-minute time resolution at the Institut für Umweltphysik in Heidelberg (Germany, 49°25'N, 8°41'E, 116 m a.s.l +30 m a.g.l.) (see Fig. B1 for $CO_2$ and $\delta^{13}C(CO_2)$). A description of the measurement principle can be found in Esler et al. (2000) and Griffith et al. (2010, 2012). Hammer et al. (2012) describe the Heidelberg-specific instrumental set-up in detail and Vardag et al. (2015a) describe modifications to this set-up and the calibration strategy for the stable isotopologue measurements.

The intermediate measurement precision of the FTIR is about 0.05 ppm for $CO_2$ and 0.04 ‰ for $\delta^{13}C(CO_2)$ (both 9 minute averages) as determined from the variation of daily target gas measurements (Vardag et al., 2014; Vardag et al., 2015a). In this work, we only use hourly $CO_2$ and $\delta^{13}C(CO_2)$ values, since simulation runs often have an hourly resolution and thus, observations and simulations can directly be compared. However, from Allan standard deviation tests, we know that the intermediate measurement precision of hourly measurements is only slightly better than for nine-minutely measurements (Vardag et al., 2015a).

### B2   Four years of concurrent $CO_2$ and $\delta^{13}C(CO_2)$ measurements in Heidelberg

The $CO_2$ concentration in Heidelberg varies over the course of the year and has its maximum in winter and its minimum in summer (Fig. B1). This pattern is mainly driven by larger fossil fuel emissions in winter than in summer. Especially, emissions from residential heating are higher in the cold season. Furthermore, biospheric uptake of $CO_2$ is lower in winter than in summer. The minimum of the isotopic $\delta^{13}C(CO_2)$ value coincides with the maximum in $CO_2$ concentration and vice versa. The features are anti-correlated since almost all $CO_2$ sources in the catchment area of Heidelberg are more $\delta^{13}C$-depleted than the background concentration and therefore a $CO_2$ increase always leads to a depletion of $\delta^{13}C(CO_2)$ in atmospheric $CO_2$. Also, the biospheric $CO_2$ sink, dominating in summer, discriminates against $\delta^{13}C(CO_2)$, leaving the atmosphere enriched in $^{13}C(CO_2)$, while $CO_2$ decreases. On top of the seasonal cycle, $CO_2$ in Heidelberg (Fig. B1) slightly increases over the course



of four years by about 2 ppm year$^{-1}$. At the same time $\delta^{13}$C(CO$_2$) decreases by about 0.04 ‰ year$^{-1}$. These rates are similar to the CO$_2$ increase and $\delta^{13}$C(CO$_2$) decrease rates in Mauna Loa, Hawaii, USA (Dlugokencky et al., 2015; White et al., 2015) and therefore reflect the global increase of CO$_2$ from $^{13}$C-depleted sources moderated by air-sea gas exchange. It is not visible to the eye, how the degree of depletion in $\delta^{13}$C(CO$_2$) varies over the course of the year (see Fig. B1). To analyze this behavior, the mean source signature must be computed (see Sect. 2.2 and Fig. 4).

*Author contributions.* S. Vardag developed the running Keeling approach in exchange with I. Levin. S.Vardag verified this approach using pseudo data from the STILT model and applied the approach to measured data. The measured data was partly taken by S. Hammer (until Sept. 2011) and mainly by S. Vardag (Sept. 2011 to June 2015). The final discussion and manuscript writing profited from input from all three authors.

*Acknowledgements.* This work has been funded by the InGOS EU project (284274) and national ICOS project funded by the German Ministry of Education and Research (Contract number: 01LK1225A). We thank NOAA/ESRL and INSTAAR for making their observational data from Mace Head and Mauna Loa available on the WDCGG website. Further, we acknowledge the financial support given by Deutsche Forschungsgemeinschaft and Ruprecht-Karls-Universität Heidelberg within the funding program Open Access Publishing.



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



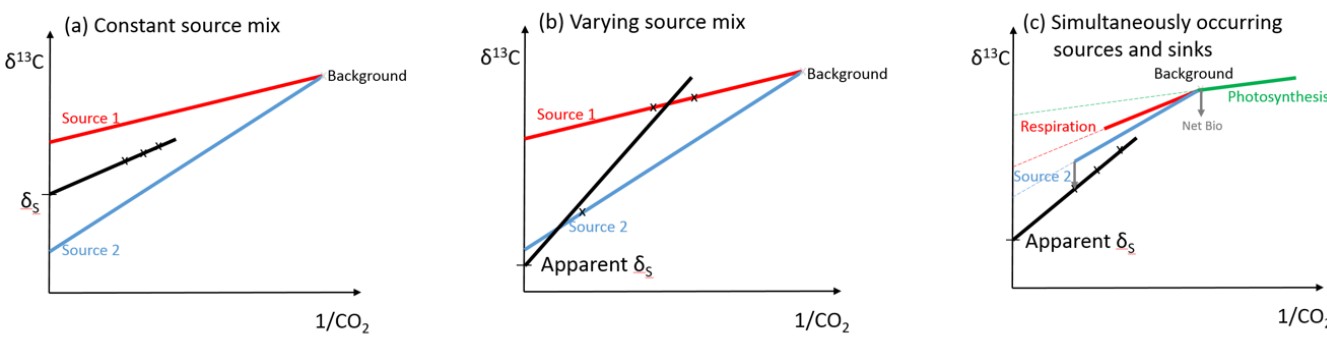

**Figure 1.** Regression-based determination of source signature using a Keeling plot. For clarity of illustration, we only draw three data points instead of five, which we use for our computation. a) Constant source mix during the time of source signature determination leads to the correct isotopic signature, $\delta_S$. b) Change of source mix during the period of determination of a Keeling plot due to either a temporal change of emission characteristics or a wind direction change leads to a biased result. These situations can be usually identified by a large error of the intercept, $\delta_S$ (we choose an error >2 ‰ to reject these results) c) Sources and sinks with different isotopic signatures or sink fractionation occur at the same time and lead to a wrong apparent source signature. Strong biases are prevented by choosing a minimum net $CO_2$ concentration range of 5 ppm and demanding a monotonous increase of $CO_2$ during the five hours (see text for more details). Note that the background value is displayed for illustration, but it is not used in the running Keeling plot approach.



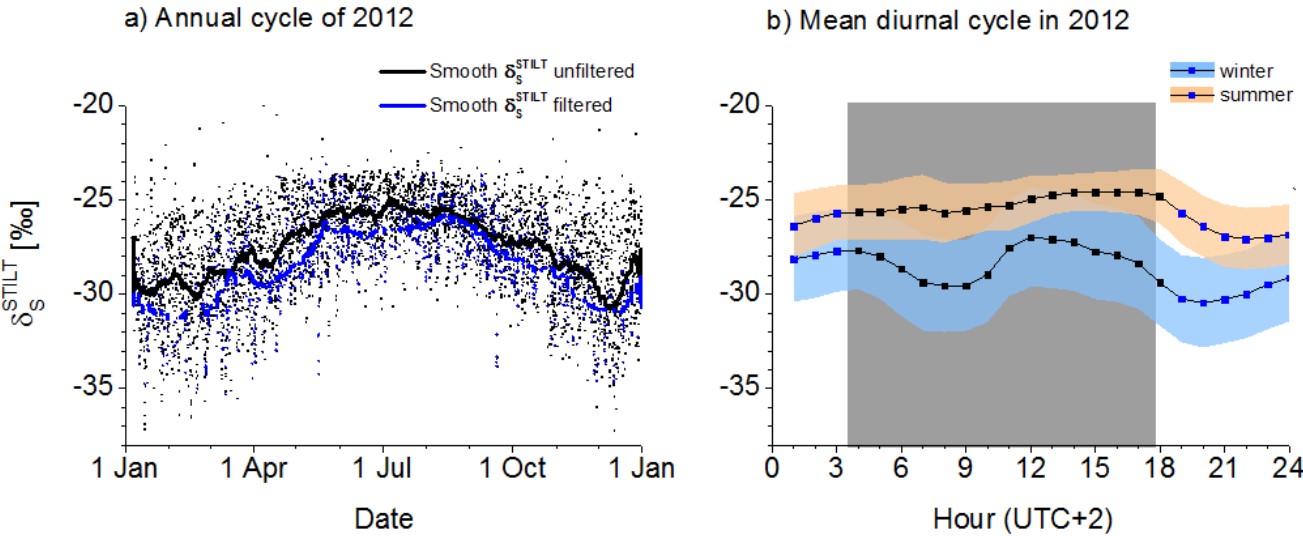

**Figure 2.** Source signature as calculated with the STILT model following equation A1. a) Unfiltered in black and filtered (for monotonous increase and minimal range) in blue. Only about 15% of all data points fulfill our strict criteria. However, they are distributed approximately evenly throughout the year. b) Diurnal cycle of modeled mean source signature due to diurnally varying mean source mix. Gray areas denote times when source signature is usually filtered out.



**Figure 3.** Comparison between modeled reference source signature (blue) and the running Keeling intercept (red), which is regression-based using the modeled $CO_2$ and $\delta^{13}C(CO_2)$ records. a) Long term comparison for the year 2012. The smoothed lines of window size 100 are also shown in the respective colors. b) Summer excerpt and c) winter excerpt (grey areas in a) of both reference and regression-based source signature. The crosses denote unfiltered data and bold stars denote filtered data.



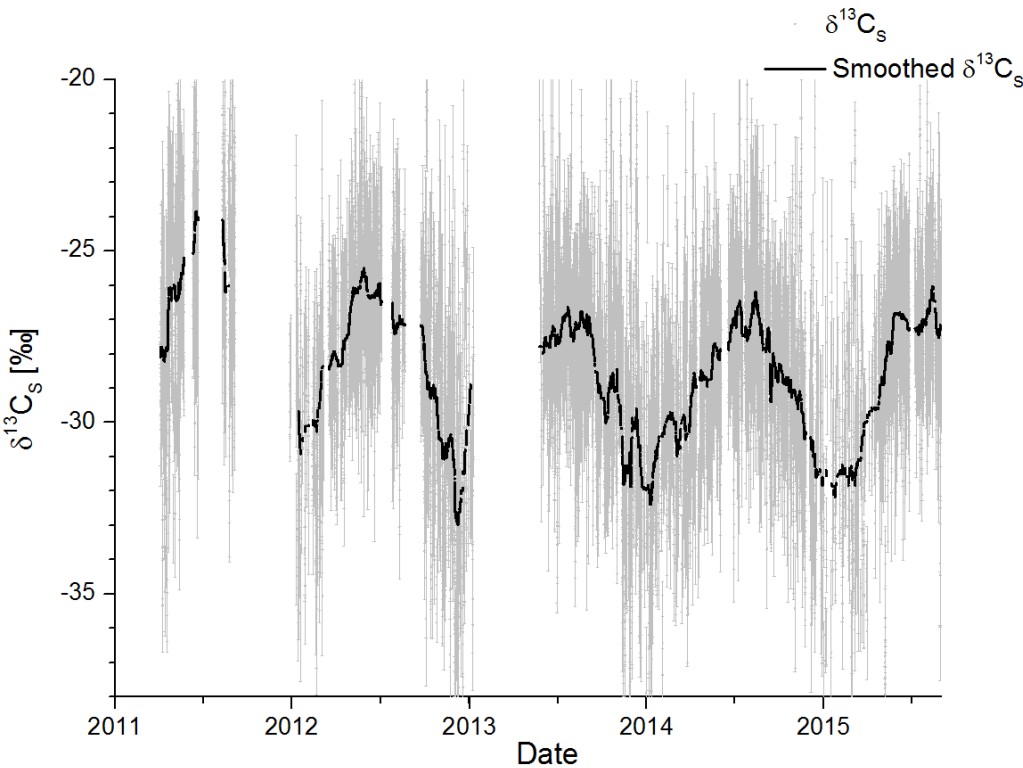

**Figure 4.** Running Keeling approach-based source signature in Heidelberg from 2011 until mid of 2015. The black line is the smoothed running Keeling signature (50%-percentile filter with window size=100 hours). Half a window size before the beginning of a large data gap the data is not further smoothed to prevent smoothing artifacts.




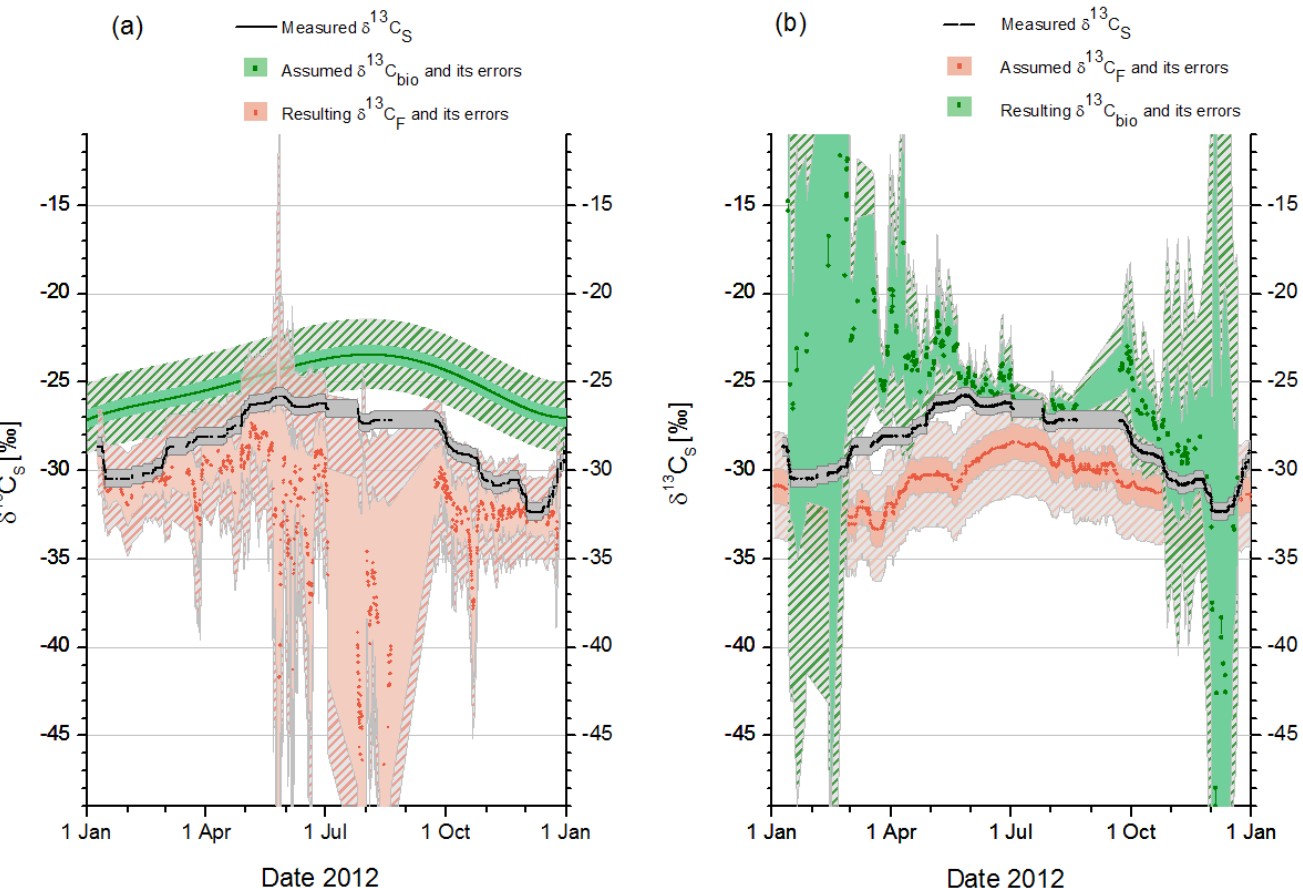

**Figure 5.** a) A fixed isotopic end member of the biosphere (green, $\pm$ uncertainty of 0.5 ‰ (light green area) and 2 ‰ (crosshatched green)) together with the measured source signature (black) results in $\delta_F$ (red, $\pm$ its uncertainty). b) A fixed isotopic end member of the fuel mix (red, $\pm$ uncertainty of 1 ‰ (salmon pink) and 2 ‰ (crosshatched gray-pink)) together with the measured source signature (black) results in $\delta_{bio}$ (green, $\pm$ its uncertainty). In both cases, also the fuel $CO_2$ share (or biospheric $CO_2$ share) is required. We here use the share calculated with STILT on the basis of EDGAR v4.3 and assume an absolute uncertainty of 10%.



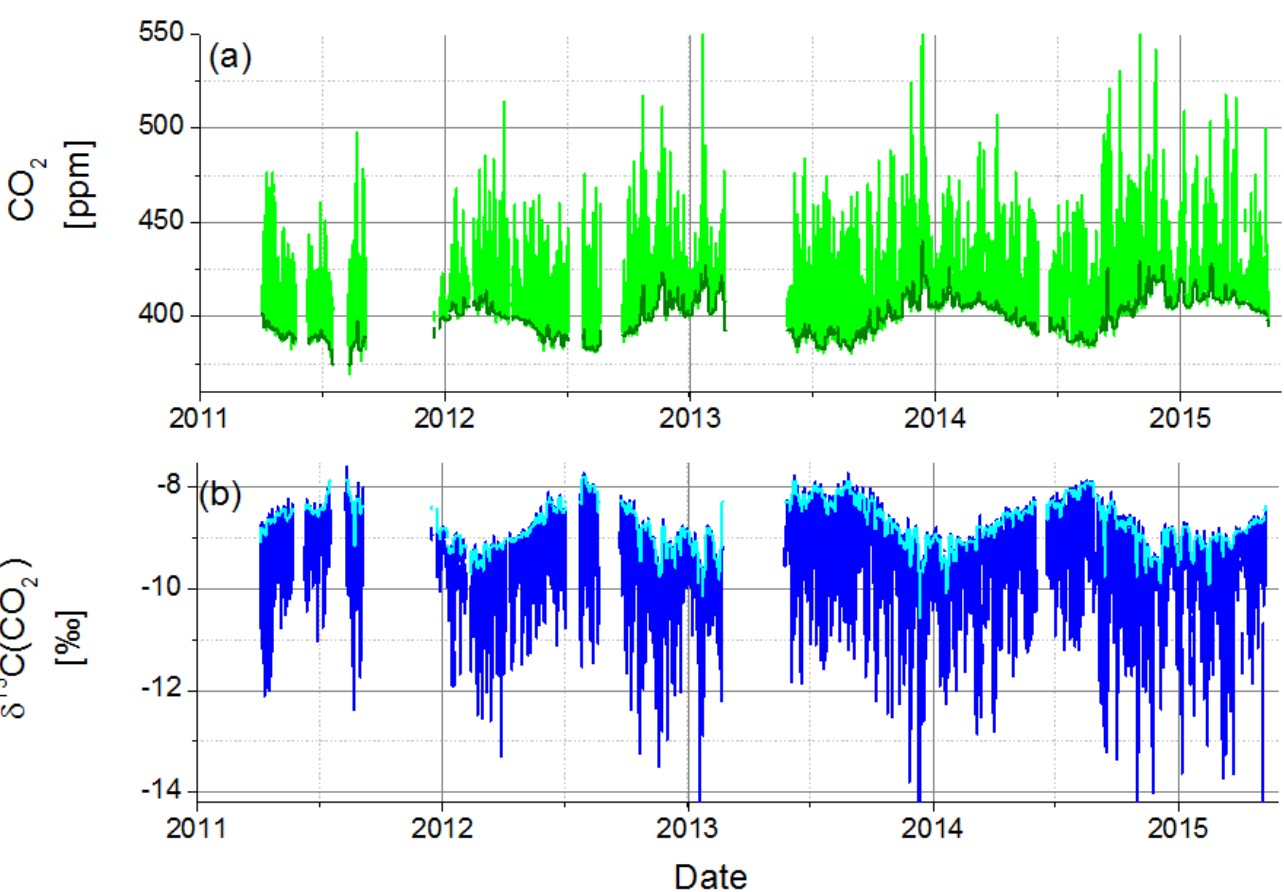

**Figure B1.** Continuous Heidelberg hourly FTIR record of (a) $CO_2$ and (b) $\delta^{13}C(CO_2)$ from April 2011- June 2015. Data gaps occur when the instrument was away during a measurement campaign or when instrumental problems occurred. The lower (and upper) 5% envelope is drawn for $CO_2$ and $\delta^{13}C(CO_2)$ in dark green and light blue, respectively.





**Table 1.** $\delta^{13}C(CO_2)$ source signature of fuel types and biosphere as used in the model and the range of literature values. Note, that for a specified region, the range of possible isotopic signature can often be narrowed down, if the origin and/or production process of the fuel type is known.

| Emission source | Used $\delta_{F,i}$ or $\delta_{bio}$ [‰] | Range of literature values $\delta_{F,i}$ or $\delta_{bio}$ [‰] | Reference |
|---|---|---|---|
| **Fuel types** | | | |
| Coal | | -23 to -27 | Mook, 2000 |
| - Hard Coal | -25 | | |
| - Brown coal | -27 | | |
| Peat | -28 | -22 to -29 | Mook, 2000; Schumacher et al., 2011 |
| Oil | -29 | -19 to -35 | Andres et al., 1994; Mook, 2000; Schumacher et al., 2011 |
| Gas | | | |
| -Natural gas | -46 | -20 to -100 | Andres et al., 1994 |
| -Derived gas | -28 | -26 to -29 | Bush et al., 2007 |
| Solid waste | -28 | -20 to -30 | typical range of C3 and C4 plant mixes (Mook, 2000) |
| Solid biomass | -27 | -20 to -30 | typical range of C3 and C4 plant mixes (Mook, 2000) |
| Bio liquid | -29 | -20 to -30 | typical range of C3 and C4 plant mixes (Mook, 2000) |
| Biogas | -11 | 0 to -16 | Widory et al., 2012; Levin et al., 1993 |
| **Biosphere** | | -20 to -30 | Lloyd and Farquhar, 1994; Mook, 2000 |
| Photosynthesis | -23 | -20 to -30 | typical range of C3 and C4 plant mixes Mook, 2000 |
| Respiration | -25 | -20 to -30 | typical range of C3 and C4 plant mixes (Mook, 2000) |