# Peer review of "Evaluation of four years continuous $\delta^{13}\text{C}(\text{CO}_2)$ data using a moving Keeling plot method"

_Biogeosciences, 2016_

## Referee Comment (RC1) · Anonymous Referee #1 · 1 Apr 2016

Review of "Evaluation of four years continuous $\delta^{13}$C(CO$_2$) data using a running Keeling approach" by S. N. Vardag, S. Hammer and I. Levin

**General:**

The manuscript deals with a four year combined record of $\delta^{13}$C(CO$_2$) and CO$_2$ from Heidelberg in Germany. An of running Keeling plot approach has been applied in order to estimate the source signatures from the data. The approach including their set criteria were tested using a STILT model dataset representing the Heidelberg conditions as good as possible. The agreement between the known source signature in these modelled dataset and those retrieved from it using their running Keeling approach is surprisingly well. The application of their approach to the four years observed dataset yields a clear seasonality of the retrieved source signature between quite well defined limits using a 100 hours smoothing filter. Then they discussed the shortcomings of the method to disentangle the different unknowns, namely the fossil fuel share and its isotope composition as well as the isotope signature of biosphere source. They conclude that it is only possible to retrieve robust results under quite strict conditions, i.e. (i) a monotonous CO$_2$ increase of at least five ppm over a five hours interval and (ii) an uncertainty of below two permil for the source signature. This restricts their derived source signature dataset by 85%, which is very substantial, which is somewhat a disadvantage. Furthermore, they nicely document that the biosphere source signal can only reliably be estimated during summer. The fossil fuel source signature is in contrast only reliable during winter, when only $\delta^{13}$C(CO$_2$) and CO$_2$ measurements are available.

**I really enjoyed reading this manuscript and I suggest accepting it with only minor revisions.**

Detailed comments:

Abstract:

L4: …opening the door to the quantification of CO$_2$ shares … or opening the door to quantify CO$_2$ shares …

L8: Disentangling this seasonal source signature into shares of source components is, however, ….

L13: …, such as D14C(CO2) or oxygen/carbon dioxide concentration ratios.

Main text

P2, L6-7: style, two times insight into ….reformulate one

P2, L32-33: eq. 2 and 3 are equivalent, therefore the about equal has to be changed to an equal sign in eq. 3.

P3, L10ff and L23ff is referring to the same topic, namely what kind of regression analyses should be used. These two parts should be combined. I personally would move the second part up.

P3, L 20f: This statement is two strict and has not been mentioned like this by Miller and Tans (2003). Otherwise, the comparison between regression filtered and STILT filtered source estimates would not be as good since most of the time simultaneous occurring sinks and sources are present.

P3, L25: What is WTLS? Is it the same as geometric mean regression (GMR) as discussed in Zobitz?

P4, L1: occurring

P4, L4: this approach leads to a strong auto-correlation of the source signature values.

P4, L5: maybe reformulate to something like: We choose five hours as a compromise between maximal number of data points and source mix constancy.

P4, L21: …as a decrease would be due …(delete of)

P4, L19ff: Why do you not apply a simple r2 criteria? Your criteria yield a significant reduction of data and corresponds to r2 larger than 0.9. What is the benefit of using your criteria of source signal uncertainty? R2 would also be independent on the regression method applied, the retrieved slopes and intercepts not. Maybe the errors are again independent, I have not checked it.

Section 2.3: (structure)

For the reader it would better to improve the visibility of the actual criteria in use: maybe with (i) … (ii)

P5, L11: …are 0-2‰ more enriched than the "filtered" source signatures (blue) as expected from our criteria.

P5, L16: …Keeling method and the used filter criteria on the model…

P5, L26: about instead of ca.?

P6, L12f: this finding is in excellent agreement with a previous source seasonality estimate by Sturm et al, which should be mentioned

Sturm, P., M. Leuenberger, F.L. Valentino, B. Lehmann, and B. Ihly, Measurements of CO2, its stable isotopes, O-2/N-2, and Rn-222 at Bern, Switzerland, *Atmospheric Chemistry and Physics*, *6*, 1991-2004, 2006.

P6, L27 delete sub-title 4.2.1

P7, L17: maybe it is better to use whether instead of if

P7, L24: …the mean measured isotope signature.

P7, L25: delete significantly

P8, L29 and 33: Why are the values different (1.5‰ and 1‰)?

P9, L2: Assuming constant isotopic end members over the course of one year, we would be able….

P9, L6: ….to the change in the fraction of respiration…

P9, L6: is it correct to say that in principle photosynthesis would also lead to an isotopic change but since you are analysing only positive CO2 gradients, i.e. CO2 release, you restricted it to respiration only. You might state this explicitly.

P9, L14: …into the fuel CO2 share.

P9, L17: ..there is a need of either …at the sources.

P9, L15ff (4.2.5): Nothing is said about the possibility to use oxygen measurement. A clear distinction between biospheric and fossil fuel sources can be calculated based on the different oxidation ratios for these two sources. Furthermore, calibrated CO/CO2 measurements are helpful as well as already documented in various studies.

P9, L27: ..aiming at an improved quantitative …

P9, L29: …and CO2 records for a potential partitioning of source contributions.

P9, L29f: this last sentence is not clear, please reformulate or delete it.

P10, L26: ditto as P8, L29 and L33.

Appendix A:

P11, L3: …air parcel originated from.

P11, eq. A1: still not clear to me why one has to use absolute concentration values. It leads to different delta values.

P18, Fig1c: The lengths of the red and green arrows is not the same since one has to balance CO2 and not 1/CO2. However, it might be not visible

P18, Caption: … or wind direction change (transportation)

P18; Caption, line 3: what do you mean with correct isotope signature, it is still a mixture and it has not been split up yet.

P20, It would be worthwhile to have the CO2 changes along with these graphs (at least for b and c.

P21: high values in 2011? Correct or artefact due to calibration issues?

P21: It would be nice to add the modelled curve for the year 2012.

P22: Why don't you use the radiocarbon that you have available and base your fossil fuel on inventory estimates?

P23: are the lower and upper 5% important? Have you used this filtering?

---

## Referee Comment (RC2) · Anonymous Referee #2 · 15 Apr 2016

This manuscript of Vardag *et al.* presents an analytical approach to evaluate the $CO_2$ source signature $\delta^{13}C_S$, using continuous, high resolution time-series of $CO_2$ and $\delta^{13}C$, recorded with an FTIR. The analysis is based on the Keeling-plot method, where a time-window of 5 hours is continuously moved across the whole data set, resulting in a continuous source signature estimate over the observation period of four-years. The manuscript is generally well written, uses an outstanding data-record and validates the proposed method using pseudo data from the STILT model. However, the major findings, like the strong limitation of the Keeling-plot method for urban catchment areas with multiple and variable sources as well as the seasonal variation of the source signatures are known since many years and discussed in a vast number of publications, some of which are also referenced by the authors. Although, it is useful (but not novel) to see the difficulties of estimating the year-round $CO_2$ fossil fuel or biosphere share in urban atmosphere using the $CO_2$ and $\delta^{13}C$ data only, the reader is left with vague alternatives and a method, which is empirically tuned to a specific spatial and temporal setting, rejecting about 85% of the estimated values. This manuscript would strongly gain scientific value by including further tracers such as $^{14}CO_2$, $^{18}CO_2$, CO, and $^{222}$Rn, discussing the advantages and pitfalls of such a combined approach, and deducing measurement strategies for future monitoring activities. As the authors have the above mentioned data (see e.g. Vogel *et al.* Tellus, 65, 2013) and a detailed model investigation (Vardag *et al*, ACP, 15, 2015), I strongly recommend using these in a concerted fashion to facilitate a better and clearer understanding of the limiting factors, requirements and identification of best practice for an efficient and unbiased monitoring of $CO_2$ source signatures. Without such major revision, the manuscript does not fulfill the high standards required for publication in Biogeosciences.

**General comments:**

A more appropriate title should be given. A "running Keeling approach" is awkward. First, the terminology broadly accepted by the community is the "Keeling plot approach (or method)". Second, the mathematical operation applied in the described approach is a moving average or moving time window. In addition, the method does not differ (except the trace gas species and window size) from the method published by Röckmann *et al.*, so I strongly recommend to not increase the number of nomenclatures unnecessarily and stick with the name of "moving Keeling plot method" as proposed by Röckmann *et al.*

If the authors write four-years in the title then they should also give the signatures for all these years and not only limit to one particular year. Otherwise, give a reason why this year was selected as representative case and give estimates how the findings for 2012 can be extended to other years.

The abstract should also reflect the major drawbacks of the method: 85% of the data are rejected, because they do not fulfil the filtering criteria, mainly night-time periods are considered, and the selected criteria are empirical and specific to a particular urban area. Furthermore, an additional smoothing (100 h window) is applied to the estimated values.

The manuscript would greatly benefit from a more conventional structure, such as Introduction, Methods, Results and Discussion. Several sub-sub-sections are not necessary and hinder the text flow, e.g. by adding many cross-references. More specifically, I recommend merging the subsections 3.1 and 3.2 into section 3 as paragraphs. Similarly, sub-subsections 4.2.1 – 4.2.5 can be included in the main text using simple paragraph-spacing.

The averaging window was selected to be 5 hours, but the motivation is weak. In principle, the FTIR is able to produce 9 minute averaged values, so why not include the resulting 33 data points into the Keeling-plot intercept determination? The higher temporal resolution should lead to a more robust fit, and a better insight into the dynamics of source signature variations, which could eventually be used as a more objective filtering instead of the empirical criteria. Just consider Figure 1 with 10 fold better resolution. Arguing with the model resolution of 1 hour is not appropriate in this context. Similarly, the argument of being a period in which the source-mix does not change significantly is ambiguous because the large amount of rejected source signature estimates. For the reader it would be very useful to learn about the optimal temporal resolution but the respective limitation of the model and the instrument does, unfortunately, not allow to draw the corresponding conclusions.

How representative are the STILT model data for urban areas? A city with its complex network of buildings and street canyons generates turbulent flows at scales that are certainly beyond the resolution of STILT. Also, what is the model sensitivity at various sampling heights within an urban area?

The filter criteria used in the manuscript are mainly fulfilled for nighttime, so it would be good to know the uncertainty of the transport model for nocturnal data.

Advection and vertical mixing can significantly influence the urban $CO_2$ signal, leading to vertical gradients. Therefore, wind speed and direction data are most likely needed to adequately interpret the observed $CO_2$ values. Thus, a discussion about the representativeness and sensitivity of the sampling site to wind speed and direction as well as its location and height would be highly recommended.

The isotopic source signature of the biosphere is found to be more depleted than previously published value, but the analysis in the present work is mainly based on nighttime data, where photosynthesis is negligible and respiration dominates. Furthermore, distinguishing between respiration, coal burning and gasoline is difficult, because they have similar $\delta^{13}C$. The authors should discuss this potential bias on their $\delta_{bio}$ estimates. For such situations, the oxygen isotope ratio ($\delta^{18}O$) could be used to distinguish between biogenic and anthropogenic $CO_2$ as the evaporative enrichment of $H_2^{18}O$ in plants and soils imparts a unique signature. At the observed regional scale, it should be possible to provide the necessary model input.

In the same context, even the pseudo data shown in Fig 2a indicate a systematic bias for the summer period between the filtered and unfiltered cases. This discrepancy should be discussed in terms of influence in determining source signatures.

The source signature value (-32.5‰) found in this work is significantly different from the value (-25‰) published by the same authors for the same year (Vardag *et al*, 2015a). A discussion about this discrepancy is required.

**Specific comments:**

Abstract, L5: "without introducing biases" is a very strong statement and probably not applicable. "reducing biases" would be more appropriate.
Abstract, L6: state which model.

Abstract, L7: are these bias values for the model data? If so, state this explicitly.

Abstract, L13: This statement should be much more quantitative, which implies significant additional information and possibly research in the main section of the paper.

Pg2, L1: use plural for optical techniques, since there are various approaches available on the market.

Pg2, L2: thereby

Pg2, L21: "bias-free", see remark above

Pg2, L27: the "classical" is not necessary, because up to date there is only this method.

Pg3, L4: this sentence is awkward, I recommend reformulating it.

Pg3, Eq3: revise the formula, the $CO_2bg$ has a positive sign.

Pg3, L14: "the Keeling plot" instead of "a Keeling plot".

Pg3, L28: why not to use measured data to test the different fit models? There should be no reason for synthetic data to deliver different results when applying different forms of the linear fitting routines. The situation can though be different when using real data.

Pg3, L29: for the very same criteria statement another reference is used (Sect 2.2 instead Sect. 2.3., see Pg3, L14)

Pg3, L30: specify, how the weights are determined?

Pg3, L31: revise the section name (see comment above regarding title)

Pg3, L33: "running" Keeling approach, again see above and delete this sentence.

Pg4, L19: The threshold criterion of 2‰ error has no objective motivation. Try to give its meaning in the context of some quantity like a confidence interval or in terms of source allocation error.

Pg4, L21: check wording "as a decrease of would be"

PG4, L28: how does this compare with a situation of 6 hour period and 4 or 6 ppm increase criteria? Is there a way to generalize these filter criteria?

Pg5. L7: give a reference for the STILT model.

Pg5. L24: what was the decision criterion for smoothing the source signatures with 100 hours window size? Evaluating the smoothing effect on pseudo data and assuming its validity on real data can be prone to errors.

Pg6, L11. Remove "Heidelberg" before "$CO_2$".

Pg6, L16. The explanation of outliers is weak and hard to understand. What do you mean by "statistical"? The filtering criteria were selected to be rather strict, so what else determines the uncertainty of the method?

Pg6, L18: are the values for inter-quartile ranges are for the smoothed data?

Pg7, L18 replace "we ask here, if we can" with "the question is whether it is possible to"

Pg9, L1. this section has nothing to do with accuracy evaluation, being more a qualitative description of various scenarios. Revision is recommended. See also suggestion above regarding text-flow.

Pg9, L15. This section is basically a repetition of what was already mentioned previously.

Pg10, L12: replace "real measured data set in Heidelberg" with "real data set measured in Heidelberg"

Fig.3 add the measured $\delta^{13}C_S$ to the figures.

Fig.5 it is somehow strange that if one considers the periods between January-April and October-December, where the measured $\delta^{13}C_S$ and assumed (or estimated) $\delta^{13}C_F$ show little deviation for both scenarios, the $\delta^{13}C_{bio}$ exhibits extreme fluctuations (Fig.5b). Furthermore, the fact that the agreement is good between model and observed $\delta^{13}C_S$ data would imply that

the summer period should look similar for the $\delta^{13}C_{bio}$ as well. In other words, what would the situation look like, when fixing both end members $\delta^{13}C_{bio}$ and $\delta^{13}C_F$, and estimating $\delta^{13}C_S$?

Appendix A, L7: *Röckmann et al.* found that fossil-fuel related emissions may be overestimated in EDGAR and using this inventory data leads to source signatures that are too enriched. Would this also apply to the $CO_2$ data presented in this work?.

Appendix A, L17-18: To what extent are the remote measurements made at Mace Head representative as background values for quantifying the regional atmospheric impact of urban $CO_2$ emissions in Heidelberg?

---

## Author Comment (AC1) · 27 Jun 2016

**Response to anonymous referee #1 on "Evaluation of four years continuous d13C(CO2) data using a running Keeling approach"**

We want to thank this anonymous reviewer for very helpful suggestions and comments, which have helped to improve the manuscript. We have revised the manuscript and outline the changes in the following.

General:

The manuscript deals with a four year combined record of $\delta_{13}C(CO_2)$ and $CO_2$ from Heidelberg in Germany. An of running Keeling plot approach has been applied in order to estimate the source signatures from the data.

The approach including their set criteria were tested using a STILT model dataset representing the Heidelberg conditions as good as possible. The agreement between the known source signature in these modelled dataset and those retrieved from it using their running Keeling approach is surprisingly well. The application of their approach to the four years observed dataset yields a clear seasonality of the retrieved source signature between quite well defined limits using a 100 hours smoothing filter. Then they discussed the shortcomings of the method to disentangle the different unknowns, namely the fossil fuel share and its isotope composition as well as the isotope signature of biosphere source. They conclude that it is only possible to retrieve robust results under quite strict conditions, i.e. (i) a monotonous $CO_2$ increase of at least five ppm over a five hours interval and (ii) an uncertainty of below two permil for the source signature. This restricts their derived source signature dataset by 85%, which is very substantial, which is somewhat a disadvantage.

Furthermore, they nicely document that the biosphere source signal can only reliably be estimated during summer. The fossil fuel source signature is in contrast only reliable during winter, when only $\delta_{13}C(CO_2)$ and $CO_2$ measurements are available. I really enjoyed reading this manuscript and I suggest accepting it with only minor revisions.

Detailed comments:

Abstract: L 4: ...opening the door to the quantification of $CO_2$ shares...or opening the door to quantify $CO_2$ shares...

We have added "the", such that it reads: "…opening the door to the quantification of…".

L8: Disentangling this seasonal source signature into shares of source components is, however, ....

We have changed this in the revised manuscript.

L13: ..., such as D14C(CO2) or oxygen/carbon dioxide concentration ratios.

As oxygen/carbon dioxide concentration ratios have not been used quantitatively to distinguish between fossil fuel and biospheric $CO_2$, we have decided to not include this sentence here in the abstract. For consistency, we also remove $^{14}C(CO_2)$ in the abstract. However, we mention both tracer methods in the conclusion of the revised manuscript instead.

Main text
P2, L6-7: style, two times insight into ....reformulate one

We have changed the second one to "This may be used to study biospheric responses…".

P2, L32-33: eq. 2 and 3 are equivalent, therefore the about equal has to be changed to an equal sign in eq. 3.

We use an equal sign in the revised manuscript.

P3, L10ff and L23ff is referring to the same topic, namely what kind of regression analyses should be used. These two parts should be combined. I personally would move the second part up.

L10ff refers to the difference between Keeling plot approach and Miller-Tans plot approach and l. 23ff refers to the fitting algorithm. We agree that these two topics should be discussed together and move the second part up, as suggested by the reviewer.

P3, L 20f: This statement is two strict and has not been mentioned like this by Miller and Tans (2003). Otherwise, the comparison between regression filtered and STILT filtered source estimates would not be as good since most of the time simultaneous occurring sinks and sources are present.

Miller and Tans (2003) state that "counter-intuitive results can occur any time fluxes of opposing sign are combined and then sampled in the atmosphere" and that "the precision […] depends on choosing our measurement environment to match closely the assumptions in our models". We agree that the statement "the determination of source mix is not per se possible" is too strict and instead write that biases may be introduced when fluxes of opposing sign occur simultaneously.

P3, L25: What is WTLS? Is it the same as geometric mean regression (GMR) as discussed in Zobitz?

WTLS (weighted total least square fit) is similar to a ODR (orthogonal distance regression) used by Zobitz et al. (2006). It has been developed by Krystek and Anton (2007) as stable algorithm for line fitting and takes into account the uncertainty in x and y direction. We give the citation and add a comment in the revised manuscript.

P4, L1: occurring

We have changed this in the revised manuscript.

P4, L4: this approach leads to a strong auto-correlation of the source signature values.

We have changed this in the revised manuscript to "this approach leads to a strong auto-correlation of neighboring source signature values."

P4, L5: maybe reformulate to something like: We choose five hours as a compromise between maximal number of data points and source mix constancy.

We have reformulated this sentence to "We choose five hours as a compromise between number of data points and thus, of robust regression, and source mix constancy."

P4, L21: ...as a decrease would be due ...(delete of)

We have deleted "of".

P4, L19ff: Why do you not apply a simple r2 criteria? Your criteria yield a significant reduction of data and corresponds to r2 larger than 0.9.
What is the benefit of using your criteria of source signal uncertainty?
R2 would also be independent on the regression method applied, the retrieved slopes and intercepts not. Maybe the errors are again independent, I have not checked it.

As the reviewer points out, $R^2$ is the same for different regression models. The reason is that $R^2$ is independent of the uncertainty of $1/CO_2$ and $\delta^{13}C$. However, the standard deviation of the offset (and of the slope) in the WTLS-fit takes into account the errors in x and y (see Eqs. 24 c,d in Krystek and Anton, 2007). We therefore prefer using the standard deviation of the offset instead of $R^2$.

In preliminary work, which we do not show in the manuscript, we have also tried using a $R^2 > 0.9$ criteria to filter the data and have found no significant differences to using the standard deviation of the offset. Both criteria seem to be similarly valid in the studied catchment area.

Section 2.3:
(structure) For the reader it would better to improve the visibility of the actual criteria in use: maybe with (i) ...(ii)

We have adapted this numbering at the end of section 2.3 in the revised manuscript to summarize the filter criteria.

P5, L11: ...are 0 - 2‰ more enriched than the "filtered" source signatures (blue) as expected from our criteria.

We feel that the insertion "as expected from our criteria" is not necessary here, as this is explained in the following sentence.

P5, L16: ...Keeling method and the used filter criteria on the model...

We have changed this in the revised manuscript.

P5, L26: about instead of ca.?

We now use "about" in the revised manuscript.

P6, L12f: this finding is in excellent agreement with a previous source seasonality estimate by Sturm et al, which should be mentioned:
Sturm, P., M. Leuenberger, F.L. Valentino, B. Lehmann, and B. Ihly, Measurements of CO2, its stable isotopes, O2/N2, and Rn-222 at Bern, Switzerland, Atmospheric Chemistry and Physics, 6, 1991-2004, 2006.

We have added this reference by Sturm et al., 2006 in the revised manuscript together with Schmidt, 1999.

P6, L27 delete sub-title 4.2.1

We have deleted the subtitle 4.2.1. Note that upon suggestion of reviewer #2, we have applied a new and more classical structure to the manuscript.

P7, L17: maybe it is better to use whether instead of if

We use "whether" in the revised manuscript.

P7, L24: ...the mean measured isotope signature.

Ok.

P7, L25: delete significantly

We have deleted significantly in the revised manuscript.

P8, L29 and 33: Why are the values different (1.5 ‰ and 1‰)?

They should actually be the same and be 1.0 ‰. We have changed this in the revised manuscript.

P9, L2: Assuming constant isotopic end members over the course of one year, we would be able....

We have made this stylistic change in the revised manuscript.

P9, L6: ....to the change in the fraction of respiration...

We have added "the" in this sentence.

P9, L6: is it correct to say that in principle photosynthesis would also lead to an isotopic change but since you are analysing only positive $CO_2$ gradients, i.e. $CO_2$ release, you restricted it to respiration only. You might state this explicitly.

Yes, this is correct. We make a respective comment in the revised manuscript (Sect. 2.3).

P9, L14: ...into the fuel $CO_2$ share.

We added "share" at the end of this sentence.

P9, L17: ..there is a need of either ...at the sources.

We have deleted this subsubsection on request of reviewer #2, and have embedded it into the conclusions. There we now use "there is a need of" instead of "must be", as recommended.

P9, L15ff (4.2.5): Nothing is said about the possibility to use oxygen measurement. A clear distinction between biospheric and fossil fuel sources can be calculated based on the different oxidation ratios for these two sources. Furthermore, calibrated $CO/CO_2$ measurements are helpful as well as already documented in various studies.

In this study, we want to focus on using $\delta^{13}C$ and $CO_2$ only as tracer. However, it is true that $O_2/N_2$ measurements provide an additional promising tracer to separate between fossil fuel, biogenic and oceanic sources (e.g. Keeling, 1988; Bender et al., 2005; Steinbach et al., 2011). To our knowledge, using $O_2/N_2$ as tracer for fossil fuels on a regional scale has not been comprehensively studied so far. On the other hand, studies using $CO/CO_2$ at regional scale are various (e.g. Levin and Karstens, 2007; Vogel et al., 2010; Vardag et al., 2015b). We have added a short comment in the conclusion, but we do not ponder upon the different tracers, as this is not the scope of the manuscript.

P9, L27: ..aiming at an improved quantitative ...

We have made this stylistic change in the revised manuscript.

P9, L29: ...and CO2 records for a potential partitioning of source contributions.

We have changed this in the revised manuscript.

P9, L29f: this last sentence is not clear, please reformulate or delete it.

We have deleted this sentence in the revised manuscript.

P10, L26: ditto as P8, L29 and L33.

We change these numbers that both read 1.0 ‰ in the revised manuscript.

Appendix A: P11, L3: ...air parcel originated from.

We have added "from" in the revised manuscript.

P11, eq. A1: still not clear to me why one has to use absolute concentration values. It leads to different delta values.

Indeed using the absolute values leads to different numbers. However, as Miller and Tans (2003) point out and as we elaborate in this manuscript (see Fig. 1c), as soon as negative fluxes occur, the resulting source signature does not lie within the range of the source signature end members anymore. Respective results are, thus, not interpretable as gross-flux weighted mean source signature anymore. As we are interested in determining the gross-flux weighted mean source signature, we take absolute values for the calculation of the reference mean source signature. In this way, we can check if the computed source signature equals the gross-flux weighted mean of all sources; with that is an interpretable and intuitive measure.

P18, Fig1c: The lengths of the red and green arrows is not the same since one has to balance $CO_2$ and not $1/CO_2$. However, it might be not visible

Upon impulse of this reviewer, we have checked and can confirm that the difference is not visible for typical $CO_2$ (and $1/CO_2$) ranges.

P18, Caption: ...or wind direction change (transportation)

We have changed this in the revised manuscript.

P18; Caption, line 3: what do you mean with correct isotope signature, it is still a mixture and it has not been split up yet.

Here, we mean the flux-weighted mean isotopic source signature (following eq. A1 in the manuscript). We have added a comment in the figure caption of the revised manuscript.

P20, It would be worthwhile to have the CO2 changes along with these graphs (at least for b and c).

We add the $CO_2$ changes to Fig. 3 in the revised manuscript.

P21: high values in 2011? Correct or artefact due to calibration issues?

Even though 2011 was the very first year of our measurements, we do not have any hints (e.g. target gas measurements or other), which would explain any artefact due to calibration issues. Therefore, we suggest that this is a real effect.

P21: It would be nice to add the modelled curve for the year 2012.

We have now added the modelled curve for the year 2012 in Figure 4.

P22: Why don't you use the radiocarbon that you have available and base your fossil fuel on inventory estimates?

Good point: We have actually considered including this record initially. However, we have decided to leave $^{14}CO_2$ out because of three main reasons:
1) We felt that the main statement of the manuscript, which concerns the usefulness and pitfalls when evaluating and interpreting continuous $\delta^{13}C$-$CO_2$ measurements, would be weakened by including an additional tracer.
2) Many monitoring stations do not have $^{14}C$ measurements available. Therefore, our study is more representative if not including the $^{14}C$-$CO_2$ measurements, but using (generally available) emission inventory data instead.
3) We only have integrated samples of $^{14}C(CO_2)$ available, which cannot be compared to continuous $\delta^{13}C$-$CO_2$ measurements directly due to the integration effect (described in Vardag et al., 2015b).

P23: are the lower and upper 5% important? Have you used this filtering?

They are not used in the manuscript. We have removed these in the revised manuscript.

References used in this reply:

Bender, M. L., Ho, D. T., Hendricks, M. B., Mika, R., Battle, M. O., Tans, P. P., Conway, T.J., Sturtevant, B. & Cassar, N.: Atmospheric O2/N2 changes, 1993–2002: Implications for the partitioning of fossil fuel CO2 sequestration. Global Biogeochemical Cycles, 19(4), 2005.

Keeling, R. F.: Measuring correlations between atmospheric oxygen and carbon dioxide mole fractions: A preliminary study in urban air. Journal Of Atmospheric Chemistry, 7(2), 153-176, 1988.

Krystek, M. and Anton. M.: A weighted total least-squares algorithm for fitting a straight line. Measurement Science and Technology 18.11: 3438, 2007.

Levin, I., Karstens, U.: Inferring high-resolution fossil fuel CO2 records at continental sites from combined 14CO2 and CO observations. Tellus B 59.2: 245-250, 2007.

Miller, J. B. and Tans, P. P.: Calculating isotopic fractionation from atmospheric measurements at various scales, Tellus, pp. 207–214, 2003.

Schmidt, M.: Messung und Bilanzierung anthropogener Treibhausgase in Deutschland, Dissertation, Ruprecht-Karls-Universität Heidelberg, 1999.

Steinbach, J., Gerbig, C., Rödenbeck, C., Karstens, U., Minejima, C., & Mukai, H.: The CO2 release and Oxygen uptake from Fossil Fuel Emission Estimate (COFFEE) dataset: effects from varying oxidative ratios. Atmospheric Chemistry and Physics, 11(14), 6855-6870, 2011.

Sturm, P., Leuenberger, M., Valentino, F. L., Lehmann, B., & Ihly, B.: Measurements of CO2, its stable isotopes, O2/N2, and 222Rn at Bern, Switzerland. Atmospheric Chemistry and Physics, 6(7), 1991-2004, 2006.

Vardag, S. N., Gerbig, C., Janssens-Maenhout, G., and Levin, I.: Estimation of continuous anthropogenic CO2: model-based evaluation of CO2, CO, δ13C(CO2) and Δ14C(CO2) tracer methods, Atmospheric Chemistry and Physics, 15, 12705-12729, doi:10.5194/acp-15-12705-2015, 2015.

Vogel, F. R., Hammer, S., Steinhof, A., Kromer, B., & Levin, I.: Implication of weekly and diurnal 14C calibration on hourly estimates of CO-based fossil fuel CO2 at a moderately polluted site in southwestern Germany. Tellus B, 62(5), 512-520, 2010.

Zobitz, J., Keener, J., Schnyder, H., and Bowling, D.: Sensitivity analysis and quantification of uncertainty for isotopic mixing relationships in carbon cycle research, Elsevier-Agricultural and Forest Meteorology, 136, 2006

---

## Author Comment (AC2) · 27 Jun 2016

**Response to anonymous referee #2 on "Evaluation of four years continuous d13C(CO2) data using a running Keeling approach"**

We want to thank this anonymous reviewer for very helpful suggestions and comments. We have revised the manuscript respectively and outline our replies and changes in the following.

This manuscript of Vardag et al. presents an analytical approach to evaluate the CO2 source signature $\delta^{13}C_S$, using continuous, high resolution time-series of CO2 and $\delta^{13}C$, recorded with an FTIR. The analysis is based on the Keeling-plot method, where a time-window of 5 hours is continuously moved across the whole data set, resulting in a continuous source signature estimate over the observation period of four-years. The manuscript is generally well written, uses an outstanding data-record and validates the proposed method using pseudo data from the STILT model. However, the major findings, like the strong limitation of the Keeling-plot method for urban catchment areas with multiple and variable sources as well as the seasonal variation of the source signatures are known since many years and discussed in a vast number of publications, some of which are also referenced by the authors. Although, it is useful (but not novel) to see the difficulties of estimating the year-round CO2 fossil fuel or biosphere share in urban atmosphere using the CO2 and δ13C data only, the reader is left with vague alternatives and a method, which is empirically tuned to a specific spatial and temporal setting, rejecting about 85% of the estimated values. This manuscript would strongly gain scientific value by including further tracers such as 14CO2, 18CO2, CO, and 222Rn, discussing the advantages and pitfalls of such a combined approach, and deducing measurement strategies for future monitoring activities. As the authors have the above mentioned data (see e.g. Vogel et al. Tellus, 65, 2013) and a detailed model investigation (Vardag et al, ACP, 15, 2015), I strongly recommend using these in a concerted fashion to facilitate a better and clearer understanding of the limiting factors, requirements and identification of best practice for an efficient and unbiased monitoring of CO2 source signatures. Without such major revision, the manuscript does not fulfill the high standards required for publication in Biogeosciences.

A lot of instruments measuring atmospheric $\delta^{13}C(CO_2)$ continuously have been installed recently with the objective of better understanding the measured $CO_2$ signal (e.g. Torn et al., 2011; Tuszon et al., 2011; Griffith et al., 2012; Griffis, 2013; Sturm et al., 2013; Vardag et al., 2015b). The expectation when using these continuously measuring instruments was to disentangle different $CO_2$ source contributions at high temporal resolution

and with that, to obtain a complete picture of the source mixes and their variations at different measurement sites.

Many studies have shown qualitatively how a $CO_2$ and $\delta^{13}C$ record could be used (e.g. Zimnoch et al., 2010; Tuszon et al., 2011; van Asperen et al., 2014; Moore and Jacobson, 2015; Newman et al., 2016), paving the way towards a more comprehensive understanding of the $CO_2$ record in different settings.

However, to our knowledge, no study has yet calculated the mean $CO_2$ source signature at high temporal resolution (hourly) over a period of more than a year in an urban setting. Moreover, this is the first comprehensive evaluation showing that the retrieved source signature is not biased. This bias-check using synthetic data is vital, especially for $CO_2$, because the prerequisites of the Keeling plot method need to be fulfilled in order to obtain correct results.

Our paper aims at showing 1) how the source signature can be obtained from a continuous $\delta^{13}C$ and $CO_2$ record 2) what can be learned from a continuous isotopic source signature record alone and 3) where additional information is needed. To our knowledge, these aspects have not been discussed elsewhere, but we would highly appreciate hints on which work we might have overlooked.

Concerning the technical part, we share the reviewer's disappointment about the need to reject 85% of the data. However, this seems to be an intrinsic problem for an urban setting with multiple sources and sinks: Obviously, in many situations the prerequisites of the Keeling plot method are not fulfilled at our site. Thus, rejecting 85% of the data seems inevitable if biases in the source signature shall be minimized. We understand that we failed in making this point clear and have elaborated this important finding in more detail in the revised manuscript.

Further, the actual percentage of rejected data points depends on the biases, which are tolerated by the data user. But it depends also and especially on the setting (wind direction change, number of sources in the catchment area, photosynthetic flux, etc.). Therefore, we are not able to provide a universal recipe how to calculate the source signature that would be applicable at every given setting. We rather demonstrate how one can check the obtained mean source signature and which parameters indicate potential biases in the mean source signature.

Concerning the interpretation of the mean source signature, the results of this paper might seem sobering to some readers, as we failed to provide

here a straight-forward way to estimate the fossil fuel component using only $\delta^{13}C(CO_2)$ and $CO_2$. However, we feel that it is important and timely to clearly state what can be learned and what cannot be learned from combined $\delta^{13}C(CO_2)$ and $CO_2$ measurements alone. This manuscript may, therefore, also guide the decision whether or not it is worth to equip a measurement station with $CO_2$ and $\delta^{13}C$ instruments.

As the reviewer states, the limitations of $\delta^{13}C(CO_2)$ -based approaches are often mentioned in publications. However, the consequences of these limitations are generally not discussed thoroughly, and results are presented without stressing the many assumptions, which are needed to obtain a quantitative result. We do not want to follow the approach to "fix" the problem by using more tracers plus additional assumptions as this was already done in different studies. E.g., Vardag et al. (2015b) compared different tracers ($CO_2$, $\delta^{13}C(CO_2)$, CO and $^{14}C$) and tracer combinations to estimate the fossil fuel share, as well as possible calibration strategies to obtain the stable isotope end members and with that the fossil fuel share using $^{14}C$. They have discussed advantages and pitfalls of combined approaches and have deduced specific measurement strategies for future monitoring activities.

In the present manuscript, we follow a more puristic approach by using only $\delta^{13}C(CO_2)$ and $CO_2$ measurements as firstly, many measurement stations do not have additional tracers available and secondly to highlight the additional assumptions required for a quantitative year-round determination of $CO_2$ source signatures.

Furthermore, we have considered including $^{18}O-CO_2$ in this study, as suggested by the reviewer. However, we are very certain that the $^{18}O-CO_2$ record will not provide additional insight into the isotopic signature of the fossil fuel sources or the plant respiration signal, at least not without implementing additional water isotope measurements and a sophisticated carbon-water model, as $^{18}O-CO_2$ is strongly coupled to the $^{18}O-H_2O$ signal (see Vardag et al., 2015a). Even though the usage of the $^{18}O-CO_2$ record is in general a very interesting field of research, it by far exceeds the scope of this paper.

Moreover, it has been shown that $^{222}Rn$ can be used to distinguish between concentration changes due to changes of the planetary boundary layer height and concentration changes due to emissions (Levin et al., 1999). This is especially valuable when deriving emissions from concentration changes (Schmidt et al., 2001). However, the source

signature itself is independent of the absolute $CO_2$ signal. It only depends on the relative fossil fuel and biospheric $CO_2$ shares. Thus, the calculated source signature is independent of atmospheric mixing conditions. Therefore, we feel that it is not helpful to use $^{222}$Rn in this study.

Finally, we want to ascertain that we clearly see (and discuss in the manuscript) the shortfalls of using $\delta^{13}C(CO_2)$ and $CO_2$ only, as is correctly pointed out by the reviewer. However, demonstrating these shortfalls and at the same time showing the usefulness of collocated continuous $CO_2$ and $\delta^{13}C(CO_2)$ records, is exactly what we attempt to show in this paper and what we stress in the revised manuscript.

General comments:
A more appropriate title should be given. A "running Keeling approach" is awkward. First, the terminology broadly accepted by the community is the "Keeling plot approach (or method)". Second, the mathematical operation applied in the described approach is a moving average or moving time window. In addition, the method does not differ (except the trace gas species and window size) from the method published by Röckmann et al., so I strongly recommend to not increase the number of nomenclatures unnecessarily and stick with the name of "moving Keeling plot method" as proposed by Röckmann et al.

We agree that it may be helpful to follow the nomenclature of Röckmann et al.; we thus changed "running Keeling approach" to "moving Keeling plot method" in the title and throughout the entire manuscript, as suggested by the reviewer.

If the authors write four-years in the title then they should also give the signatures for all these years and not only limit to one particular year. Otherwise, give a reason why this year was selected as representative case and give estimates how the findings for 2012 can be extended to other years.

The mean source signature was computed for four years (see Fig. 4 of the manuscript). However, we do the analysis of the end members $\delta_{bio}$ and $\delta_F$ only for the year 2012 for two reasons. Firstly, in order to demonstrate the wealth of information from $\delta^{13}C(CO_2)$ and $CO_2$ only, we feel that it suffices to analyze only one year. Secondly, as described in the manuscript, the fuel contributions of the emission inventory (EDGAR) are only available for 2010 and have already been extrapolated to the year 2012. Therefore, for the years 2011-2015, we would have to use the same source mix as for

the year 2012, providing no additional insight. We add a respective comment to the revised manuscript (Sect. 3.3).

The abstract should also reflect the major drawbacks of the method: 85% of the data are rejected, because they do not fulfil the filtering criteria, mainly nighttime periods are considered, and the selected criteria are empirical and specific to a particular urban area. Furthermore, an additional smoothing (100 h window) is applied to the estimated values.

The reason why we reject 85% of the data is intrinsic as a Keeling plot can only be performed in situations, which fulfill the basic assumptions of the Keeling plot. We have added a respective comment in the abstract and explain this in more detail in the manuscript, as we seem to have failed to make this point clear in the original version of the manuscript.
The smoothing is applied in order to expose the synoptic and seasonal trends in the figures.

The manuscript would greatly benefit from a more conventional structure, such as Introduction, Methods, Results and Discussion. Several sub-sub-sections are not necessary and hinder the text flow, e.g. by adding many cross-references. More specifically, I recommend merging
the subsections 3.1 and 3.2 into section 3 as paragraphs. Similarly, sub-subsections 4.2.1 – 4.2.5 can be included in the main text using simple paragraph-spacing.

We have adopted a more conventional structure such that chapter 2 was named "Methods", chapter 3 "Results and Discussion" and chapter 4 "Summary and Conclusions". Also we have removed all subsubsections and structure the revised manuscript by subsections only.

The averaging window was selected to be 5 hours, but the motivation is weak. In principle, the FTIR is able to produce 9 minute averaged values, so why not include the resulting 33 data points into the Keeling-plot intercept determination? The higher temporal resolution should lead to a more robust fit, and a better insight into the dynamics of source signature variations, which could eventually be used as a more objective filtering instead of the empirical criteria. Just consider Figure 1 with 10 fold better resolution. Arguing with the model resolution of 1 hour is not appropriate in this context. Similarly, the argument of being a period in which the source-mix does not change significantly is ambiguous because the large amount of rejected source signature estimates. For the reader it would be very useful to learn about the optimal temporal resolution but the respective limitation of the model and the instrument does, unfortunately, not allow to draw the corresponding conclusions.

This is a very good point. The FTIR is even able to produce 3-minute data points. Nevertheless, we have decided to use a 5 hour moving window for Keeling plot determination using hourly $CO_2$ and $\delta^{13}C(CO_2)$ data, which we discuss and explain in the following.

As the reviewer states correctly, our motivation for taking hourly data points comes from the model, which is available only at hourly resolution. An important output of our study is that we can assure that we are determining the source signature correctly. This can only be done by comparing it with the known source signature, which is provided by the model. We find this check not only appropriate, but also rather essential (and novel). Therefore, we disagree with the reviewer and in contrary understand that arguing with the model resolution of 1 hour is indeed appropriate.

As the reviewer points out, for the determination of the mean source signature using measured data, it may be advantageous to use 3-minutely instead of hourly values. In this case, changes of the source signature within an hour can also contribute to the scattering of the fit, improving or deteriorating the fit, but potentially providing additional information.
However, we have no means (as the model resolution is not high enough) to check whether we get correct results when taking three-minutely values.

For curiosity and upon impulse of reviewer #2, we have nonetheless checked if the Keeling plot gives different results when using 3-minutely measurements instead of hourly averaged measurements. We have therefore calculated the source signature for the entire year 2012 in a 5 hour moving window using 3 minutely data (100 data points) and using hourly data (5 data points), but applying the same filter criteria (standard deviation of the offset <2 ‰, $CO_2$ increase >5ppm). We actually find a (small) differences between the source signatures at similar: The mean difference between the Keeling plot intercept using 3 minutely data and the Keeling plot intercept using hourly data (both in an 5 hour moving window) is 0.2 +/- 1.3 ‰ (non smoothed). It is not possible to answer where the difference comes from and if the results still give the gross-flux weighted mean source signature, as it cannot be compared to any reference (provided by the model). Therefore, we are obliged to use hourly instead of three-minutely values for computation of the Keeling plot.

A second point raised by reviewer #2 concerns the length of the moving window. We have experimented using smaller (1h, 3h, 4h) and larger (6h, 7h, 8h) averaging windows and compared how much of the data is rejected by the filter criteria for which window size. For the one hour window size, we filtered data with $CO_2$ increase less than 1 ppm over one hour and standard deviation of the offset > 2 ‰, and for five hours, we

rejected data with $CO_2$ increase less than 5 ppm and standard deviation > 2 ‰ and respectively for the other window sizes. We found that for a window size of 5 hours, the coverage is maximal (ca. 15%) showing that five hours is the compromise between a period in which we encounter a significant increase of $CO_2$ (at fixed uncertainties of the $CO_2$ and $\delta^{13}C(CO_2)$ data), but also a period in which the source mix remains more or less constant and $CO_2$ is still increasing. For comparison, when using a window size of only one hour, the coverage is only ca. 5%. We have now added a comment in the revised manuscript (Sect. 2.2) to provide the reader with a more objective criteria for choosing the correct window size.

How representative are the STILT model data for urban areas? A city with its complex network of buildings and street canyons generates turbulent flows at scales that are certainly beyond the resolution of STILT. Also, what is the model sensitivity at various sampling heights within an urban area?

It is not of utmost importance that the STILT model is absolutely correct. We use the STILT $CO_2$ and $\delta^{13}C(CO_2)$ data set to determine the isotopic source signature and compare it to the modelled reference source signature. For this consistency check, it is important that the source mix is realistic, but it does not need to be exactly correct.
The modelled planetary boundary layer height introduces a large biases into the modelled concentration. However, as already mentioned above, the mean source signature is independent of the absolute concentrations, but only depends on the $CO_2$ shares. Therefore, as a rough indicator of the different variability of the source mix in the model and the measurements, we have compared the interquartile ranges of the mean source signatures around the smoothed mean source signature. It is 1.2 ‰ for model data and 1.8 ‰ for measured data (see Sect. 3.2), indicating a similar, but slightly lower (30%) variability in the model than in the measurements. We have added a respective comment in the revised manuscript.

The sampling height used in the model equals the actual measurement sampling height. As we are only interested in evaluating the data from this sampling height, we do not feel that it is necessary to elaborate the sensitivity for other sampling heights.

The filter criteria used in the manuscript are mainly fulfilled for nighttime, so it would be good to know the uncertainty of the transport model for nocturnal data. Advection and vertical mixing can significantly influence the urban CO2 signal, leading to vertical gradients. Therefore, wind speed

and direction data are most likely needed to adequately interpret the observed CO2 values. Thus, a discussion about the representativeness and sensitivity of the sampling site to wind speed and direction as well as its location and height would be highly recommended.

The STILT model has a transport error of about 40% during the daytime and up to 100% at night (Gerbig et al., 2008). This is mainly due to the uncertainty of the planetary boundary layer height affecting all absolute concentrations. However, the absolute value of the concentration does not influence the mean source signature. Only the share of the different components change the mean source signature. Therefore, the STILT model can be used to test the moving Keeling plot method, despite large vertical transport errors.

As the reviewer states correctly, the measured signal at the measurement station is strongly influenced by the footprint of the measurement site, which itself is influenced by advection, vertical mixing, wind speed, wind direction etc.. It is not the scope of this manuscript to make a detailed footprint analysis. We focus here only on the ability to derive the source signatures of fossil fuel emitters and the biosphere irrespective of what footprint we are looking at. We therefore do not discuss the measurement site and meteorological parameters in detail in the revised manuscript, but provide a reference for the Heidelberg measurement site and the catchment area (Vogel et al. (2010)).

The isotopic source signature of the biosphere is found to be more depleted than previously published value, but the analysis in the present work is mainly based on nighttime data, where photosynthesis is negligible and respiration dominates. Furthermore, distinguishing between respiration, coal burning and gasoline is difficult, because they have similar $\delta^{13}C$. The authors should discuss this potential bias on their $\delta$bio estimates. For such situations, the oxygen isotope ratio ($\delta$18O) could be used to distinguish between biogenic and anthropogenic CO2 as the evaporative enrichment of H218O in plants and soils imparts a unique signature. At the observed regional scale, it should be possible to provide the necessary model input.

We are not sure, which published value the reviewer is referring to. To our knowledge the literature values are distributed around -25 ‰ (+/-2 ‰) (see e.g. Mook, 2001; Ballantyne et al., 2011 and others). As there seems to be hardly any discrimination during respiration (Lin et al., 1997), the nighttime respiration values should not differ significantly from daytime respiration values; therefore, the assumed biospheric isotopic signature is

our best estimate. The potential bias of the biospheric source signature end member is discussed in former Sect. 4.2.3 (now: Sect. 3.4) and is illustratively demonstrated in Fig. 5a where we take into account two different possible uncertainties to demonstrate the effect of the biospheric end member.

The reviewer correctly points out that distinguishing between respiration and coal burning is difficult, since they have a similar $\delta^{13}C(CO_2)$ value. That of gasoline is slightly more depleted, but still rather close, which makes a clear distinction of these sectors difficult. However, a distinction between the mean fossil fuel source and mean respiration source is still possible, as natural gas contribution leads to a more depleted mean fossil fuel signature.

We do not find it useful to include the oxygen isotope ratio to distinguish between biogenic and anthropogenic $CO_2$. As stated above, the reason is that the $H_2^{18}O$ signal (and consequently also $CO^{18}O$ signal) is highly variable, not well-known and different for soils and plants. Further, additional effects as e.g. soil invasion flux needs to be taken into account before using $\delta^{18}O$ quantitatively (see Vardag et al., 2015a). All these influences on the $^{18}O\text{-}CO_2$ signal need to be modelled with a coupled carbon-water model, which is fed by high-resolution meteorological data (e.g. precipitation, temperature etc.) and isotopic $H_2O$ data. Such a model and such measurements would be very interesting to have, but are not available in Heidelberg and at many other stations.

In the same context, even the pseudo data shown in Fig. 2a indicate a systematic bias for the summer period between the filtered and unfiltered cases. This discrepancy should be discussed in terms of influence in determining source signatures.

The reviewer correctly remarks that the pseudo data is more depleted after filtering, as daytime data is more likely to be filtered out (see also Sect. 3.1). In the Conclusion we clearly state that the long-term source signature is only representative of the nighttime. As this point is very important, we now explicitly state in the conclusions of the revised manuscript that, obviously, as a consequence, also the isotopic end members of biospheric and fossil $CO_2$ can only be estimated in periods where the mean source signature can be computed, excluding especially daytime periods.

The source signature value (-32.5‰) found in this work is significantly different from the value (-25‰) published by the same authors for the same year (Vardag et al, 2015a). A discussion about this discrepancy is required.

The reviewer probably refers to Vardag et al. (2015b), where the mean biospheric value is -25 ‰. This is in correspondence to the present study, where the mean biospheric value is -25 ‰ as well, but the mean fossil fuel signature is -32.5 ‰. In Vardag et al. (2015a), the different fossil fuel sources are further separated into traffic, residential heating, energy production etc., but again the isotopic values in no way conflict the isotopic signatures used or found in the present publication.

Specific comments:
Abstract, L5: "without introducing biases" is a very strong statement and probably not applicable. "reducing biases" would be more appropriate.

In the revised manuscript we have deleted "without introducing biases" in that sentence and use "minimal biases" in the next (new) sentence.

Abstract, L6: state which model.

We explicitly name the model here.

Abstract, L7: are these bias values for the model data? If so, state this explicitly.

These are given for the model, as we are not able to quantify the bias for real data. We added this in the abstract.

Abstract, L13: This statement should be much more quantitative, which implies significant additional information and possibly research in the main section of the paper.

As elaborated in the first passage of this reply, we have aimed to demonstrate how much information can be retrieved by using $\delta^{13}C(CO_2)$ and $CO_2$ only. Other papers have dealt with a combination of different tracers (e.g. Vardag et al., 2015b), but this is not the scope of this paper. Even though some readers might find this result devastating, we think that it is important and novel to state the advantages and shortcomings in all explicitness.

Pg2, L1: use plural for optical techniques, since there are various approaches available on the market.

We have changed this to plural in the revised manuscript.

Pg2, L2: thereby

Ok.

Pg2, L21: "bias-free", see remark above

We have changed this to "retrieval with minimal biases".

Pg2, L27: the "classical" is not necessary, because up to date there is only this method.

We have removed classical.

Pg3, L4: this sentence is awkward, I recommend reformulating it.

We have reformulated this in the revised manuscript.

Pg3, Eq3: revise the formula, the CO2bg has a positive sign.

We have checked the formula and Eq. 3 seems to be appropriate as it is, but Eq. 4, must have a minus in front of $CO_{2bg}$. We correct this in the revised manuscript.

Pg3, L14: "the Keeling plot" instead of "a Keeling plot".

We have changed this in the revised manuscript.

Pg3, L28: why not to use measured data to test the different fit models? There should be no reason for synthetic data to deliver different results when applying different forms of the linear fitting routines. The situation can though be different when using real data.

As we added a statistical noise to the synthetic data (representing measurement uncertainty), the comparison of the linear fitting routines should not differ when using synthetic data or measured data. We therefore feel that it is not necessary to repeat this analysis with measured data.

Pg3, L29: for the very same criteria statement another reference is used (Sect 2.2 instead Sect. 2.3., see Pg3, L14)

Section 2.3 is the one, which we want to refer to in both cases. We have corrected this in the revised manuscript.

Pg3, L30: specify, how the weights are determined?

In the WTLS-fit, the uncertainties in x and y direction are both used to calculate the weights. In the revised manuscript we provide a citation to the WTLS-fit (Krystek and Anton, 2007) explaining also the weights (Eq. 11 in Krystek and Anton, 2007).

Pg3, L31: revise the section name (see comment above regarding title)

We have changed this in the entire manuscript.

Pg3, L33: "running" Keeling approach, again see above and delete this sentence.

Ok.

Pg4, L19: The threshold criterion of 2‰ error has no objective motivation. Try to give its meaning in the context of some quantity like a confidence interval or in terms of source allocation error.

The criteria have been motivated theoretically (see Fig. 1 and Sect. 2.3), but, as the reviewer points out correctly, the absolute values of these filter criteria have been established empirically (Sect. 2.3).
However, we check the effectiveness of the filter criteria by using the synthetic data set. This model comparison (difference between the model reference source signature and the Keeling plot based source signature of synthetic data) provides the basis for choosing the filter criteria and the interquartile range of the difference provides a measure of the precision of the estimate.
Nevertheless, for other measurement stations, other filter criteria would apply, depending on how heterogeneous the sources in the catchment area are and how fast the footprint changes. Therefore, it is not possible to provide a universal recipe for other measurement stations, but we only demonstrate how to choose the filter criteria based on the model comparison. We have added a comment about the generalization of these filter criteria in the revised manuscript (Sect. 2.3).

Pg4, L21: check wording "as a decrease of would be"

We have changed this in the revised manuscript.

PG4, L28: how does this compare with a situation of 6 hour period and 4 or 6 ppm increase criteria? Is there a way to generalize these filter criteria?

We choose a 5 hour moving window, as a compromise between a period in which we encounter a significant increase of $CO_2$ (at fixed uncertainties of the $CO_2$ and $\delta^{13}C(CO_2)$ data), but also a period in which the source mix remains constant and the $CO_2$ is still increasing. When using a window size of 5 hours, the coverage is maximal (ca. 15%) showing that 5 hours is preferential over other window sizes. For 6 and 4 hours moving windows, the coverage is slightly smaller than for 5 hours, but the source signature is not significantly different from using a 5 hour average.

Again, we are aware that our filter criteria were chosen empirically (after consulting the differences to the modelled source signature). This means, we have chosen our filter criteria such that the Keeling plot method prerequisites are fulfilled for our observational setting. This is necessary in order to obtain correct results.
Ideally, it would be nice to generalize the filter criteria and provide a universal recipe how to filter the source signature for every possible setting. However, as pointed out before, each measurement site is unique, has different absolute $CO_2$ variations, different emission patterns and footprint changes; therefore, we are unfortunately (but unavoidably) unable to generalize the filter.

What we can do and what we discuss in the manuscript (last paragraph in Sect. 3.1) is how a loosening of the filter criteria (higher or lower $CO_2$ increase) affects the data coverage and the biases introduced. We chose meaningful and descriptive scenarios so that the reader gets a feeling for the importance of the filter criteria.

Pg5. L7: give a reference for the STILT model.

We have added the reference (Lin et al., 2003) for the STILT model.

Pg5. L24: what was the decision criterion for smoothing the source signatures with 100 hours window size? Evaluating the smoothing effect on pseudo data and assuming its validity on real data can be prone to errors.

We have chosen the window size of n=100 hours (ca. 4 days), so that the synoptical and seasonal variations of the mean source signature can be

seen independent from diurnal variation. We have commented on this in the revised manuscript (Sect. 3.1).
As the simulated and real mean source signature nearly show the same variability, uncertainty and coverage, the smoothing effect of both data sets should not be biasing.

Pg6, L11. Remove "Heidelberg" before "CO2".

Ok.

Pg6, L16. The explanation of outliers is weak and hard to understand. What do you mean by "statistical"? The filtering criteria were selected to be rather strict, so what else determines the uncertainty of the method?

The pseudo-data experiment showed that even though we apply rather strict filter criteria, there are some outliers, which lead to an interquartile range of 1.2 ‰ (see Sect. 3.2). The interquartile range of all measured data points (around the smoothed curve) is 1.8 ‰ and with that only slightly higher than what we obtained from the pseudo-data, showing that the outliers in the source signature are not unusual for our applied filter criteria. We have reformulated the sentence.

Pg6, L18: are the values for inter-quartile ranges are for the smoothed data?

No, they are for the hourly non-smoothed data. We clearly point this out in the revised manuscript.

Pg7, L18 replace "we ask here, if we can" with "the question is whether it is possible to"

Ok.

Pg9, L1. this section has nothing to do with accuracy evaluation, being more a qualitative description of various scenarios. Revision is recommended. See also suggestion above regarding text-flow.

We agree that the term accuracy might not be well chosen as we discuss the resulting absolute values of the isotopic signatures rather than the quality of the data. We have therefore removed this title and instead discuss the content of this paragraph in Sect. 3.4 of the revised

manuscript. Note that we also have shortened this section, and instead discuss the implications of this section in the conclusions.

Pg9, L15. This section is basically a repetition of what was already mentioned previously.

We deleted this paragraph in the revised manuscript and fed the information to the "Summary and Conclusion" chapter instead, where it is more appropriate to summarize the findings.

Pg10, L12: replace "real measured data set in Heidelberg" with "real data set measured in Heidelberg".

We have changed this in the revised manuscript.

Fig.3 add the measured δ13CS to the figures.

Instead of adding the measured $\delta^{13}C_S$ in Fig. 3, we have decided to add the modelled $\delta^{13}C_S$ in Fig.4, so that both can be compared. This was suggested by Reviewer #1.

Fig.5 it is somehow strange that if one considers the periods between January-April and October-December, where the measured δ13CS and assumed (or estimated) δ13CF show little deviation for both scenarios, the δ13Cbio exhibits extreme fluctuations (Fig.5b). Furthermore, the fact that the agreement is good between model and observed δ13CS data would imply that the summer period should look similar for the δ13Cbio as well. In other words, what would the situation look like, when fixing both end members δ13Cbio and δ13CF, and estimating δ13CS?

The fluctuation of the estimated biospheric isotopic end member is high in winter as the biospheric $CO_2$ share is low and the source signature $\delta^{13}C_S$ is close to the assumed fossil fuel end member in winter. The biospheric end member is therefore not well constrained by $\delta^{13}C_S$, leading to a large uncertainty of the biospheric end member.
Furthermore, in Fig. 5, we use only the measured (not modelled) $\delta^{13}C_S$. Thus, we are sorry, but do not understand the second part of the reviewers comment.

Appendix A, L7:
Röckmann et al. found that fossil-fuel related emissions may be overestimated in EDGAR and using this inventory data leads to source

signatures that are too enriched. Would this also apply to the CO2 data presented in this work?

If the fossil fuel share were overestimated also for $CO_2$, the mean source signature would be influenced by a too large share by fossil fuels and a too small share of the biosphere. Therefore, in order to obtain the same measured mean source signature, the resulting source signature of fossil fuels (and of the biosphere) would be too enriched.
However, since we do not have any reliable information on an overestimation of the fossil fuel $CO_2$ share in EDGAR, and as the fossil sources of $CO_2$ and $CH_4$ are very different, we do not incorporate this in the revised manuscript.

Appendix A, L17-18: To what extent are the remote measurements made at Mace Head representative as background values for quantifying the regional atmospheric impact of urban CO2 emissions in Heidelberg?

The STILT model uses TM3 boundary conditions to retrieve the $CO_2$ concentration at the model domain boundary. TM3 is fed by measurements at different clean air sites, including Mace Head. The European model domain boundary is geographically very close to Mace Head. Therefore, the STILT domain value is closely follows the measurements performed at Mace Head for $CO_2$, which we were able to confirm for two exemplary years (not shown). In this manuscript, we use the correlation of $CO_2$ and $\delta^{13}C$ in Mace Head to achieve a boundary $\delta^{13}C$ value, which seems reasonable regarding the good agreement in $CO_2$. This boundary value is necessary to compute the modelled total $\delta^{13}C$.

However, for the moving Keeling plot approach presented in this study, we do not require an explicit background value (see Sect. 2.2) and therefore the choice of the boundary $\delta^{13}C$ hardly influences the resulting source signature.

References used in this reply:

Ballantyne, A. P., Miller, J. B., Baker, I. T., Tans, P. P., and White, J. W. C.: Novel applications of carbon isotopes in atmospheric $CO_2$: what can atmospheric measurements teach us about processes in the biosphere?, Biogeosciences, 8, 3093–3106, doi:10.5194/bg-8-3093-2011, 2011.

Gerbig, C., Lin, J. C., Wofsy, S. C., Daube, B. C., Andrews, A. E., Stephens, B. B., Bakwin, P. S., and Grainger, C. A.: Toward constraining

regional-scale fluxes of $CO_2$ with atmospheric observations over a continent: 2. Analysis of COBRA data using a receptor-oriented framework, J. Geophys. Res.-Atmos., 108,4756, doi:10.1029/2003JD003770, 2003

Gerbig, C., Körner, S., and Lin, J. C.: Vertical mixing in atmospheric tracer transport models: error characterization and propagation, Atmos. Chem. Phys., 8, 591–602, doi:10.5194/acp-8-591-2008, 2008.

Griffis, T. J.: Tracing the flow of carbon dioxide and water vapor between the biosphere and atmosphere: A review of optical isotope techniques and their application. Agricultural and Forest Meteorology 174: 85-109, 2013.

Griffith, D. W. T., Deutscher, N. M., Caldow, C., Kettlewell, G., Riggenbach, M., and Hammer, S.: A Fourier transform infrared trace gas and isotope analyser for atmospheric applications, Atmos. Meas. Tech., 5, 2481-2498, doi:10.5194/amt-5-2481-2012, 2012.

Krystek, M., and Anton, M.: A weighted total least-squares algorithm for fitting a straight line. Measurement Science and Technology 18.11: 3438, 2007.

Levin, I., Glatzel-Mattheier, H., Marik, T., Cuntz, M., Schmidt, M. and Worthy, D. E.: Verification of German methane emission inventories and their recent changes based on atmospheric observations. J. Geophys. Res. 104, 3447–3456, 1999.

Lin, G, and Ehleringer, J. R.: Carbon isotopic fractionation does not occur during dark respiration in C3 and C4 plants. Plant Physiology 114.1: 391-394, 1997.

Lin, J. C., Gerbig, C., Wofsy, S. C., Andrews, A. E., Daube, B. C., Davis, K. J., and Grainger, C. A.: A near-field tool for simulating the upstream influence of atmospheric observations: The Stochastic Time-Inverted Lagrangian Transport (STILT) model, J. Geophys. Res., 108, 4493, doi:10.1029/2002JD003161, 2003.

Mook, W. M. E.: Environmental Isotopes in the Hydrological Cycle. Principles and Applications, UNESCO/IAEA Series, http://www.naweb.iaea.org/napc/ih/documents/global_cycle/Environment al%20Isotopes%20in%20the%20Hydrological%20Cycle%20Vol%201.pd f, 2001.

Moore, J., & Jacobson, A. D.: Seasonally varying contributions to urban $CO_2$ in the Chicago, Illinois, USA region: Insights from a high-resolution $CO_2$ concentration and $\delta^{13}C$ record. Elementa: Science of the Anthropocene, 3(1), 000052, 2015.

Newman, S., Xu, X., Gurney, K. R., Hsu, Y. K., Li, K. F., Jiang, X., Keeling, R., Feng, S., O'Keefe, D., Patarasuk, R., Wong, K. W., Rao, P., Fischer, M. L., and Yung, Y. L.: Toward consistency between trends in bottom-up $CO_2$ emissions and top-down atmospheric measurements in the Los Angeles megacity, Atmos. Chem. Phys., 16, 3843-3863, doi:10.5194/acp-16-3843-2016, 2016.

Röckmann, T., Eyer, S., van der Veen, C., Popa, M. E., Tuzson, B., Monteil, G., Houweling, S., Harris, E., Brunner, D., Fischer, H., Zazzeri, G., Lowry, D., Nisbet, E. G., Brand, W. A., Necki, J. M., Emmenegger, L., and Mohn, J.: In-situ observations of the isotopic composition of methane at the Cabauw tall tower site, Atmospheric Chemistry and Physics Discussions, 2016, 1–43, doi:10.5194/acp-2016-60, http: //www.atmos-chem-phys-discuss.net/acp-2016-60/, 2016

Schmidt, M., Glatzel-Matteier, H., Sartorius, H., Worthy, D., and Levin, I.: Western European $N_2O$ emissions: A top-down approach based on atmospheric observations." Journal of Geophysical Research, 2001.

Sturm, P., Tuzson, B., Henne, S., and Emmenegger, L.: Tracking isotopic signatures of $CO_2$ at the high altitude site Jungfraujoch with laser spectroscopy: analytical improvements and representative results, Atmos. Meas. Tech., 6, 1659-1671, doi:10.5194/amt-6-1659-2013, 2013.

Torn, M. S., Biraud, S. C., Still, C. J., Riley, W. J., and Berry, J. A.: Seasonal and interannual variability in $^{13}C$ composition of ecosystem carbon fluxes in the U.S. Southern Great Plains, Tellus B, 63, 181–195, doi:10.1111/j.1600-0889.2010.00519.x, http://dx.doi.org/10.1111/ j.1600-0889.2010.00519.x, 2011.

Vardag, S. N., Hammer, S., Sabasch, M., Griffith, D. W. T., and Levin, I.: First continuous measurements of $\delta^{18}O$-$CO_2$ in air with a Fourier transform infrared spectrometer, Atmos. Meas. Tech., 8, 579-592, doi:10.5194/amt-8-579-2015, 2015a.

Vardag, S. N., Gerbig, C., Janssens-Maenhout, G., and Levin, I.: Estimation of continuous anthropogenic $CO_2$: model-based evaluation of $CO_2$, CO, $\delta^{13}C(CO_2)$ and $\Delta^{14}C(CO_2)$ tracer methods, Atmos. Chem. Phys., 15, 12705-12729, doi:10.5194/acp-15-12705-2015, 2015b.